# IMPROVING DOMAIN GENERALIZATION WITH DOMAIN RELATIONS

**Huaxiu Yao**[1,2*]**, Xinyu Yang**[3*]**, Xinyi Pan**[4]**, Shengchao Liu**[5]**, Pang Wei Koh**[6]**, Chelsea Finn**[1]
[1]Stanford University, [2]UNC-Chapel Hill, [3]CMU, [4]UCLA, [5]Caltech, [6]University of Washington
`huaxiu@cs.unc.edu, xinyuya2@andrew.cmu.edu, cbfinn@cs.stanford.edu`

## ABSTRACT

Distribution shift presents a significant challenge in machine learning, where models often underperform during the test stage when faced with a different distribution than the one they were trained on. This paper focuses on domain shifts, which occur when the model is applied to new domains that are different from the ones it was trained on, and propose a new approach called $D^3G$. Unlike previous methods that aim to learn a single model that is domain invariant, $D^3G$ leverages domain similarities based on domain metadata to learn domain-specific models. Concretely, $D^3G$ learns a set of training-domain-specific functions during the training stage and reweights them based on domain relations during the test stage. These domain relations can be directly obtained and learned from domain metadata. Under mild assumptions, we theoretically prove that using domain relations to reweight training-domain-specific functions achieves stronger out-of-domain generalization compared to the conventional averaging approach. Empirically, we evaluate the effectiveness of $D^3G$ using real-world datasets for tasks such as temperature regression, land use classification, and molecule-protein binding affinity prediction. Our results show that $D^3G$ consistently outperforms state-of-the-art methods.

## 1 INTRODUCTION

Distribution shift is a common problem in real-world applications (Gulrajani and Lopez-Paz, 2021; Koh et al., 2021a). When the test distribution differs from the training distribution, machine learning models often experience a significant decline in performance. In this paper, we specifically focus on addressing domain shifts, which arise when applying a trained model to new domains that differ from its training domains. An example of this is predicting how well a drug will bind to a specific target protein. In drug discovery, each protein is a specific domain (Ji et al., 2022), and the binding task on each domain is expensive due to the cost of lab experiments. Thus, an open challenge is to train a robust model that can be generalized to novel protein domains, which is essential when searching for potential drug candidates that can bind to proteins associated with newly discovered diseases.

To address domain shifts, prior domain generalization approaches mainly learn a single model that is domain invariant (Arjovsky et al., 2019; Krueger et al., 2021a; Li et al., 2018b; Sun and Saenko, 2016; Yao et al., 2022b), and differ in techniques they use to encourage invariance. These methods have shown promise, but there remains significant room for improvement under real-world domain shifts such as those in the WILDS benchmark (Koh et al., 2021b). Unlike learning a single domain-invariant model, we posit that models may perform better if they were specialized to a given domain. The advantage of learning multiple domain-specific models is that traditional domain generalization methods assume predictions are based solely on "causal" or general features. However, in practical scenarios, different domains can exhibit strong correlations with non-general features, and domain-specific models can exploit these features to make more accurate predictions. While there are clearly some possible benefits to learning domain-specific models, it remains unclear how to construct a domain-specific model for a *new* domain seen at test time, without any training data for that domain.

To resolve this challenge, we propose a novel approach called $D^3G$ to learn a set of diverse, training domain-specific functions during the training stage, where each function corresponds to a single

---

*Equal contribution. Work was done during Xinyu Yang's remote internship at Stanford.

domain. For each test domain, D³G leverages the domain relations to weight these training domain-specific functions and perform inference. Our approach is based on two main hypotheses: firstly, similar domains exhibit similar predictive functions, and secondly, the test domain shares sufficient similarities with some of the training domains. By capitalizing on these hypotheses, we can develop a robust model for each test domain that incorporates information about its relation with the training domains. These domain relations are derived from domain meta-data, such as protein-protein interactions or geographical proximity, in various applications. Additionally, D³G incorporates a consistency regularizer that utilizes training domain relations to enhance the training of domain-specific predictors, especially for data-insufficient domains.

Through our theoretical analysis under mild assumptions, we demonstrate that D³G achieves superior out-of-domain generalization by leveraging domain relations to reweight training domain-specific functions, surpassing the performance of traditional averaging methods. To further validate our findings, we conduct comprehensive empirical evaluations of D³G on diverse datasets encompassing both synthetic and real-world scenarios with natural domain shifts. The results unequivocally establish the superiority of D³G over best prior method, exhibiting an average improvement of 10.6%.

## 2 PRELIMINARIES

**Out-of-Distribution Generalization.** In this paper, we consider the problem of predicting the label $y \in \mathcal{Y}$ based on the input feature $x \in \mathcal{X}$. Given training data distributed according to $P^{tr}$, we train a model $f$ parameterized by $\theta \in \Theta$ using a loss function $\ell$. Traditional empirical risk minimization (ERM) optimizes the following objective:

$$\arg\min_{\theta \in \Theta} \mathbb{E}_{(x,y) \sim P^{tr}}[\ell(f_\theta(x), y)]. \tag{1}$$

The trained model is evaluated on a test set from a test distribution $P^{ts}$. When distribution shift occurs, the training and test distributions are different, i.e., $P^{tr} \neq P^{ts}$.

Concretely, following Koh et al. (2021b), we consider a setting in which the overall data distribution is drawn from a set of domains $\mathcal{D} = \{1, \ldots, D\}$, where each domain $d \in \mathcal{D}$ is associated with a domain-specific data distribution $P_d$ over a set $(X, Y, d) = \{(x_i, y_i, d)\}_{i=1}^{n_d}$. The training distribution and test distribution are both considered to be mixture distributions of the $D$ domains, i.e., $P^{tr} = \sum_{d \in \mathcal{D}} r_d^{tr} P_d$ and $P^{ts} = \sum_{d \in \mathcal{D}} r_d^{ts} P_d$, respectively, where $r_d^{tr}$ and $r_d^{ts}$ denote the mixture probabilities in the training set and test set, respectively. We also define the training domains and test domains as $\mathcal{D}^{tr} = \{d \in \mathcal{D} | r_d^{tr} > 0\}$ and $\mathcal{D}^{ts} = \{d \in \mathcal{D} | r_d^{ts} > 0\}$, respectively. In this paper, we consider domain shifts, where the test domains are disjoint from the training domains, i.e., $\mathcal{D}^{tr} \cap \mathcal{D}^{ts} = \emptyset$. In addition, the domain ID of training and test datapoints are available.

**Domain Relations and Domain Meta-Data.** In this study, our objective is to address domain shift by harnessing the power of domain relations, which encapsulate the similarity or relatedness between different domains. To illustrate this concept, we focus on the protein-ligand binding affinity prediction task, where each protein is treated as an individual domain. Domains are considered related if they exhibit similar protein sequences or belong to the same protein family. To formalize these domain relations, we introduce an undirected domain similarity matrix denoted as $\mathcal{A} = \{a_{ij}\}_{i,j=1}^{D}$, where each element $a_{ij}$ quantifies the strength of the relationship between domains $i$ and $j$. In this paper, we derive the domain relations by leveraging domain meta-data $\mathcal{M} = \{m_i\}_{i=1}^{D}$, which depict the distinctive properties of each domain.

## 3 LEVERAGING DOMAIN RELATIONS FOR OUT-OF-DOMAIN GENERALIZATION

We now describe the proposed method – D³G (leveraging domain distances for out-of-domain generalization). The goal of D³G is to improve out-of-domain generalization by constructing domain-specific models. During the training phase, we employ a multi-headed network architecture where each head corresponds to a specific training domain (Figure 1(a)). This allows us to learn domain-specific functions and capture the nuances of each domain. To address the challenge of limited training data in certain domains, we introduce a consistency loss that aids in training (Figure 1(c)).

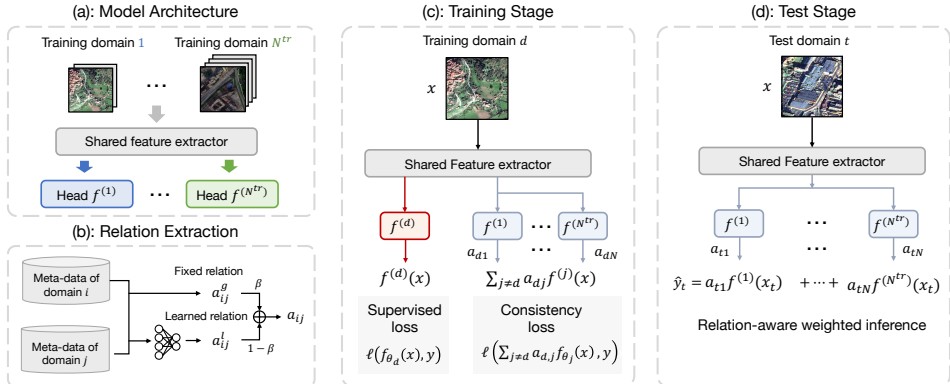

Figure 1: An illustration of D³G. (a) The multi-headed architecture of D³G, where each training domain is associated with a single head for prediction. (b) The relation extraction module, where fixed relations are extracted from domain meta-data and refined through learning from the same meta-data. (c) The training stage of D³G, where $x$ represents a single example from domain $d$, and the loss is composed of both a supervised loss and a consistency loss. (d) The test stage, where the weighting of all training domain-specific functions is used to perform inference for each test example.

During the testing phase, we construct test domain-specific models for inference by reweighing the training domain-specific models. This reweighting process takes into account the similarity between the training and test domains, allowing us to adapt the model to the specific characteristics of the test domain (Figure 1(d)). To establish domain relations, we extract information directly from domain meta-data and refine the relationships through meta-data-driven learning (Figure 1(b)). In the following sections, we delve into the details of the training and inference processes, elucidating how we construct domain-specific models and obtain domain relations.

## 3.1 BUILDING DOMAIN-SPECIFIC MODELS

In this section, we present the details of D³G for learning a collection of domain-specific functions during the training stage and leveraging these functions for relational inference during the test stage.

**Training Stage.** During the training phase, our approach utilizes a multi-headed neural network architecture comprising $N^{tr}$ heads, where $N^{tr}$ denotes the number of training domains. Given an input datapoint $(x, y)$ from domain $d$, we denote the prediction made by the $d$-th head as $f^{(d)}(x) = h^{(d)}(e(x))$, where $h^{(d)}(\cdot)$ represents the domain-specific head for domain $d$, and $e(\cdot)$ represents the feature extractor. Our objective is to minimize the predictive risk for each datapoint when using the corresponding head, ensuring accurate predictions within each domain. This is achieved by minimizing the following loss function:

$$\mathcal{L}_{pred} = \mathbb{E}_{d \in \mathcal{D}^{tr}} \mathbb{E}_{(x,y) \sim P_d} [\ell(f^{(d)}(x), y)]. \tag{2}$$

In certain scenarios, some training domains may contain limited data compared to the overall training set, posing difficulties in training domain-specific predictors. To address this challenge, we leverage the assumption that similar domains tend to have similar predictive functions. Building upon this assumption, we introduce a relation-aware consistency regularizer. For each example $(x, y)$ within a training domain $d$, the regularizer incorporates domain relations to weigh the predictions generated by all training predictors, except the corresponding predictor $f^{(d)}$. The formulation of the relation-aware consistency loss is as follows:

$$\mathcal{L}_{rel} = \mathbb{E}_{d \in \mathcal{D}^{tr}} \mathbb{E}_{(x,y) \sim P_d} \left[ \ell \left( \frac{\sum_{j=1, j \neq d}^{N^{tr}} a_{dj} f^{(j)}(x)}{\sum_{k=1, k \neq d}^{N^{tr}} a_{dk}}, y \right) \right], \tag{3}$$

where $a_{dj}$ is defined as the strength of the relation between domain $d$ and $j$. This loss encourages the groundtruth to be consistent with the weighted average prediction obtained from all other training predictors, using the domain relations to weight their contributions. By doing so, the regularizer encourages the model to: (1) rely more on predictions made by similar domains, and less on predictions made by dissimilar domains; (2) strengthen the relations between predictors and help training predictors for domains with insufficient data.

To incorporate the consistency loss into our training process, we add it to the predictive loss in equation 2 and obtain the final loss as $\mathcal{L} = \mathcal{L}_{pred} + \lambda \mathcal{L}_{rel}$, where $\lambda$ balances these two terms.

**Test Stage.** During the testing phase, D$^3$G constructs test domain-specific models based on the same assumption that similar domains have similar predictive functions. Concretely, we weight all training domain-specific functions and perform inference for each test domain $t$ by weighting the predictions from the corresponding prediction heads. Specifically, for each test datapoint $x$ drawn from the test distribution $P_t$, D$^3$G makes a prediction as follows:

$$\hat{y} = \frac{\sum_{d=1}^{N^{tr}} a_{dt} f^{(d)}(x)}{\sum_{k=1}^{N^{tr}} a_{kt}}, \tag{4}$$

where $a_{dt}$ represents the strength of the relation between the test domain $t$ and the training domain $d$. According to equation 4, for each test domain, training domains with stronger relations play a more important role in prediction. This allows D$^3$G to provide more accurate predictions by leveraging the knowledge from related domains.

## 3.2 EXTRACTING AND REFINING DOMAIN RELATIONS

We then discuss how to obtain the pairwise similarity matrix $\mathcal{A} = \{a_{ij}\}_{i,j=1}^{D}$ between different domains. Here, domain relations are derived from domain meta-data. For example, in drug-target binding affinity prediction task, where each protein is treated as a domain, we can use a protein-protein interaction network to model the relations between different proteins. Another scenario is, if we aim to predict the land use category using satelite images (Koh et al., 2021b), and each country is treated as a domain, we can use geographical proximity to model the relations among countries. The relation between domains $i$ and $j$ that is directly collected from domain meta-data is defined as $a_{ij}^{g}$. Thus, the relation between domains $i$ and $j$ is either a relation graph in domain meta-data, or the pairwise similarity calculated from each domain's meta-data. We define it as $a_{ij}^{g}$.

One potential issue with directly collecting domain relations from fixed domain meta-data is that these fixed relations may not fully reflect accurate application-specific domain relations. For example, geographical proximity can be used in any applications with spatial domain shifts, but it is hard to pre-define how strongly two nearby regions are related for different applications. To address this issue and refine the fixed relations, we propose learning the domain relations from domain meta-data using a similarity metric function. Specifically, given domain meta-data $m_i$ and $m_j$ of domains $i$ and $j$, we use a two layer neural network $g$ to learn the corresponding domain representations $g(m_i)$ and $g(m_j)$. Following Chen et al. (2020), we compute the similarity between domains $i$ and $j$ with a multi-headed similarity layer, which is formulated as follows:

$$a_{ij}^{l} = \frac{1}{R} \sum_{r=1}^{R} cos(w_r \odot g(m_i), w_r \odot g(m_j)), \tag{5}$$

where $\odot$ denotes the Hadamard product and $R$ is the number of heads. The collection of learnable weight vectors $\{w_r\}_{r=1}^{R}$ has the same dimension as the domain representation $g(m_i)$ and is used to highlight different dimensions of the vectors.

We use weighted sum to combine fixed and learned relations. Specifically, we define the relation between domains $i$ and $j$ is defined as follows:

$$a_{ij} = \beta a_{ij}^{g} + (1 - \beta) a_{ij}^{l}, \tag{6}$$

where $0 \leq \beta \leq 1$ is a hyperparameter that controls the importance of both kinds of relations. By tuning $\beta$, we can balance the contribution of the fixed and learned relations to the relation between domains. The final domain relations are used in the consistency regularization and testing stage. To summary, the pseudocodes of training and testing stages of D$^3$G is detailed in Alg. 1.

## 4 THEORETICAL ANALYSIS

In this section, we theoretically explore the underlying reasons why utilizing domain relations that are derived from domain meta-data can enhance the generalization capability to new domains. In our theoretical analysis, for an input datapoint $(x, y)$ from domain $d$, we rearrange the predictive function presented in equation 2 as $y = f^{(d)}(x) + \epsilon := h^{(d)}(e(x)) + \epsilon$, where $h^{(d)}(\cdot)$ and $e(\cdot)$ represent

---

**Algorithm 1** Training and Test Procedure of D$^3$G

---

**Require:** Training and test data, relation combining coefficient $\beta$, loss balanced coefficient $\lambda$, meta-data $\{m_d\}_{d=1}^{D}$ of all domains, learning rate $\gamma$
 1: /* *Training stage*                                                     */
 2: Initialize all learnable parameters
 3: Extract fixed relations $\{a_{ij}^g\}_{i,j=1}^{N^{tr}}$.
 4: **while** not converge **do**
 5:     Compute learned relations $\{a_{ij}^l\}_{i,j=1}^{N^{tr}}$ and obtain the final domain relations by equation 6.
 6:     **for** each example $(x, y, d)$ **do**
 7:         Calculated supervised loss $\mathcal{L}_{pred}$ by equation 2.
 8:         Computed consistency loss $\mathcal{L}_{rel}$ by equation 3 using domain relations.
 9:     Update learnable parameters with learning rate $\gamma$.
10: /* *Test stage*                                                           */
11: **for** each test domain $t$ **do**
12:     Obtain the relations between the test domain and training domains $\{a_{dt}\}_{d=1}^{N^{tr}}$
13:     **for** each example $(x, y, t)$ **do**
14:         Computed the prediction $\hat{y}$ by equation 4.

---

the head of domain $d$ and feature extractor, respectively. $\epsilon$ is a noise term which is assumed to be sub-Gaussian with a mean of 0 and a variance of $\sigma^2$.

During the testing process, we adopt the assumption that the outcome prediction function $f^{(t)}(x)$ for the test domain $t$ can be estimated using the following equation:

$$\hat{f}^{(t)}(x) = \hat{h}^{(t)}(e(x)), \text{where; } \hat{h}^{(t)} = \frac{\sum_{i=1}^{N^{tr}} a_{it}\hat{h}^{(i)}}{\sum_{k=1}^{N^{tr}} a_{kt}}, \tag{7}$$

Here, $\hat{h}^{(i)}$ represents the learned head for the training domain $i$. In the case where the denominator in equation 7 is equal to 0, we define $\hat{h}^{(t)} = 0$.

To facilitate our theoretical analysis, we make the following assumptions: (1) For each domain $d$, the domain representation $Z^{(d)}$ derived from the domain meta-data $m_d$ (i.e., $Z^{(d)} = g(m_d)$) is assumed to be uniformly distributed on $[0, 1]^r$. Furthermore, it is assumed that the domain relations accurately capture the similarity between domains. Specifically, there exists a universal constant $G$ such that for all $i, j \in \mathcal{D}$, we have $\|h^{(i)} - h^{(j)}\|_\infty \leq G \cdot \|Z^{(i)} - Z^{(j)}\|$. (2) The relation $a_{it}$ between domains $i$ and $t$ is determined by the distance between their respective domain representations and a bandwidth $B$, defined as $a_{it} = \mathbf{1}\{\|Z^{(i)} - Z^{(t)}\| < B\}$; (3) For each training domain $d$, $\hat{h}^{(d)}$ is well-learned such that $\mathbb{E}[(\hat{h}^{(d)}(e(x)) - h^{(d)}(e(x)))^2] = O(\frac{C(\mathcal{H})}{n_d})$, where $C(\mathcal{H})$ is the Rademacher complexity of the function class $\mathcal{H}$. Based on these assumptions, we then have the following theorem.

**Theorem 4.1.** *Suppose we have the number of examples $n_d \gtrsim n$ for all training domains $d \in \mathcal{D}^{tr}$, where $n$ is defined as the smallest number of examples across all domains. If the loss function $\ell$ is Lipschitz with respect to the first argument, then for the test domain $t$, the excess risk satisfies*

$$\mathbb{E}_{(x,y)\sim P_t}[\ell(\hat{f}^{(t)}(x), y)] - \mathbb{E}_{(x,y)\sim P_t}[\ell(f^{(t)}(x), y)] \lesssim B + \sqrt{\frac{C(\mathcal{H})/n}{N^{tr}B^r}}. \tag{8}$$

The theorem above implies that by considering domain relations to bridge the gap between training and test domains, the more training tasks we have, the smaller the excess risk will be. The detailed proofs are in Appendix A.1.

Building upon the results derived in Theorem 4.1, we now present a proposition that highlights the importance of obtaining a good relation matrix $A$ in enhancing out-of-domain generalization. Specifically, we compare our method with the traditional approach where all training domains are treated equally, and the similarity matrix is defined as $\tilde{A} = \{\tilde{a}_{ij}\}_{i,j=1}^{D}$, with each $\tilde{a}_{ij} = 1$. We compare the well-defined similarity matrix $A$ and ill-defined $\tilde{A}$ in the following proposition:

**Proposition 4.2.** *Under the same conditions as Theorem 4.1, suppose all $\tilde{a}_{ij} = 1$ and consider the function class $\mathcal{H} \in \{h : \|h^{(i)} - h^{(j)}\|_\infty \leq G \cdot \|Z^{(i)} - Z^{(j)}\|$ for $i, j \in \mathcal{D}\}$. Define the excess risk with similarity matrix $A$ by $R_h(\hat{f}^{(t)}, A) = \mathbb{E}_{(x,y)\sim P_t}[\ell(\hat{f}^{(t)}(x), y; A)] - \mathbb{E}_{(x,y)\sim P_t}[\ell(f^{(t)}(x), y; A)]$, we have*

$$\inf_{\hat{f}^{(t)}} \sup_{h \in \mathcal{H}} R_h(\hat{f}^{(t)}, A) < \inf_{\hat{f}^{(t)}} \sup_{h \in \mathcal{H}} R_h(\hat{f}^{(t)}, \tilde{A}). \tag{9}$$

The proposition presented above indicates that by leveraging accurate domain relations, we can achieve superior generalization performance compared to the approach of treating all training domains equally. The detailed proof of this proposition can be found in Appendix A.2.

## 5 EXPERIMENTS

In this section, we conduct a series of experiments to evaluate the effectiveness of D$^3$G. Here, we compare D$^3$G with different learning strategies and categories including (i) ERM (Vapnik, 1999), (ii) *distributionally robust optimization*: GroupDRO (Sagawa et al., 2020), (iii) *invariant learning*: IRM (Arjovsky et al., 2019), IB-IRM (Ahuja et al., 2021b), IB-ERM (Ahuja et al., 2021b), V-REx (Krueger et al., 2021a), DANN (Ganin et al., 2016b), CORAL (Sun and Saenko, 2016), MMD (Li et al., 2018a), RSC (Huang et al., 2020), CAD (Ruan et al., 2022), SelfReg (Kim et al., 2021), Mixup (Xu et al., 2020), LISA (Yao et al., 2022b), MAT (Wang et al., 2022b), (iv) *domain-specific learning*: AdaGraph (Mancini et al., 2019), RaMoE (Bui et al., 2021), mDSDI (Bui et al., 2021), AFFAR (Qin et al., 2022), GRDA (Xu et al., 2022), DRM (Zhang et al., 2022b), LLE (Li et al., 2023), DDN (Zhang et al., 2023), TRO (Qiao and Peng, 2023), . Here, methods in categories (i), (ii), (iii) learn a universal model for all domains. Additionally, for a fair comparison, we incorporate domain meta-data as features for all baselines during the training and test stages. All hyperparameters are selected via cross-validation. Detailed setups and baseline descriptions are provided in Appendix D.

### 5.1 ILLUSTRATIVE TOY TASK

**Dataset Descriptions.** Following Xu et al. (2022), we use the DG-15 dataset, a synthetic binary classification dataset with 15 domains. In each domain $d$, a two-dimensional key point $x_d = (x_{d,1}, x_{d,2})$ is randomly selected in the two-dimensional space, and the domain meta-data is represented by the angle of the point (i.e., $\arctan\left(\frac{x_{d,2}}{x_{d,1}}\right)$). 50 positive and 50 negative datapoints are generated from two Gaussian distributions $\mathcal{N}(x_d, \mathbf{I})$ and $\mathcal{N}(-x_d, \mathbf{I})$ respectively. In DG-15, we construct the fixed relations between domain $i$ and $j$ as the angle difference between key points $x_i$ and $x_j$, i.e., $a_{ij}^g = \arctan\left(\frac{x_{j,2}}{x_{j,1}}\right) - \arctan\left(\frac{x_{i,2}}{x_{i,1}}\right)$. The number of training, validation, and test domains are all set as 5. We visualize the training and test data in Figure 2a and 2b.

**Results and Analysis.** The performance of D$^3$G on the DG-15 dataset is in Figure 2. The results highlight several important findings. Firstly, it is observed that learning a single model fails to adequately capture the domain-specific information, resulting in suboptimal performance. To gain further insights into the performance improvements, Figures 2c and 2d depict the predictions from the strongest single model learning method (GroupDRO) and D$^3$G, respectively. GroupDRO learns a nearly linear decision boundary that overfits the training domains and fails to generalize on the shifted test domains. In contrast, D$^3$G effectively leverages domain meta-data, resulting in robust generalization to most test domains, with the exception of those without nearby training domains. Secondly, when compared to domain-specific learning approaches such as LLE, DDN, and TRO, D$^3$G demonstrates superior performance. This improvement can be attributed to D$^3$G's enhanced ability to capture and utilize domain relations, enabling it to achieve stronger generalization capabilities.

### 5.2 REAL WORLD DOMAIN SHIFTS

**Datasets Descriptions.** In this subsection, we briefly describe three datasets with natural distribution shifts and Appendix E provides additional details.

- **TPT-48.** TPT-48 is a weather prediction dataset, aiming to forecast the next 6 months' temperature based on the previous 6 months' temperature. Each state is treated as a domain, and the domain meta-data is defined as the geographical location. Following Xu et al. (2022), we consider two dataset splits: I. N (24) → S (24): generalizing from the 24 states in the north to the 24 states in the south; II. E (24) → W (24): generalizing from the 24 states in the east to the 24 states in the west.
- **FMoW.** The FMoW task is to predict the building or land use category based on satellite images. For each region, we use geographical information as domain meta-data. We first evaluate D$^3$G on spatial domain shifts by proposing a subset of FMoW called FMoW-Asia, including 18 countries from Asia. Then, we study the problem on the full FMoW dataset from the WILDS benchmark (Koh et al., 2021b) (FMoW-WILDS), taking into account shift over time and regions.

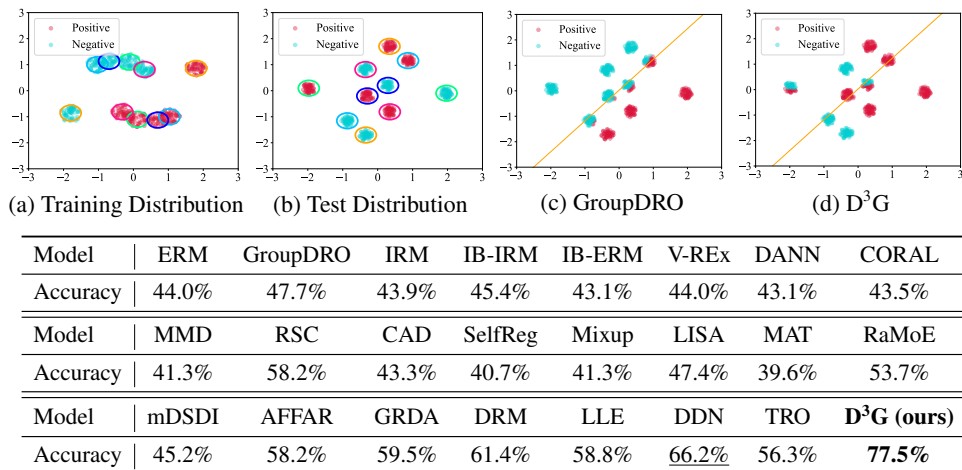

| (a) Training Distribution | (b) Test Distribution | (c) GroupDRO | (d) D$^3$G |
|---|---|---|---|

| Model | ERM | GroupDRO | IRM | IB-IRM | IB-ERM | V-REx | DANN | CORAL |
|---|---|---|---|---|---|---|---|---|
| Accuracy | 44.0% | 47.7% | 43.9% | 45.4% | 43.1% | 44.0% | 43.1% | 43.5% |

| Model | MMD | RSC | CAD | SelfReg | Mixup | LISA | MAT | RaMoE |
|---|---|---|---|---|---|---|---|---|
| Accuracy | 41.3% | 58.2% | 43.3% | 40.7% | 41.3% | 47.4% | 39.6% | 53.7% |

| Model | mDSDI | AFFAR | GRDA | DRM | LLE | DDN | TRO | D$^3$G (ours) |
|---|---|---|---|---|---|---|---|---|
| Accuracy | 45.2% | 58.2% | 59.5% | 61.4% | 58.8% | 66.2% | 56.3% | **77.5%** |

Figure 2: Results of domain shifts on toy task (DG-15). Figures (a) and (b) illustrate the training and test distributions, where datapoints in circles with the same color originate from the same domain. Figures (c) and (d) show the predicted distribution of the strongest single model method (GroupDRO) and D$^3$G. Bottom Table reports averaged accuracy over all test domains (see full table with standard deviation in Appendix F). We **bold** the best results and underline the second best results.

- **ChEMBL-STRING.** In drug discovery, we focus on molecule-protein binding affinity prediction. The ChEMBL-STRING (Liu et al., 2022) dataset provides both the binding affinity score and the corresponding domain relation. Follow Liu et al. (2022) and treat proteins and pairwise relations as nodes and edges in the relation graph, respectively. The protein-protein relations and protein structures are treated as domain meta-data. Follow Liu et al. (2022), we evaluate our method using two subsets named PPI$_{>50}$ and PPI$_{>100}$.

**Results.** We present the results of D$^3$G and other methods in Table 1. The evaluation metrics utilized in this study were selected based on the original papers that introduced these datasets (see results with more metrics in Appendix G). The findings indicate that most invariant learning approaches (e.g., IRM, CORAL) demonstrate inconsistent performance in comparison to the standard ERM. While these methods perform well on certain datasets, they underperform on others. Furthermore, even when employing domain-specific learning approaches such as LLE, DDN, and TRO, these methods exhibit inferior performance compared to learning a universal model. These outcomes suggest that these methods struggle to effectively learn accurate domain relations, even when provided with domain meta-data (more analysis in Appendix G.3). In contrast, D$^3$G, which constructs domain-specific models, achieves the best performance by accurately capturing domain relations.

## 5.3 ABLATION STUDY OF D$^3$G

In this section, we provide ablation studies on datasets with natural domain shifts to understand where the performance gains of D$^3$G come from.

**Does consistency regularization improve performance?** We analyze the impact of domain-aware consistency regularization. In Figure 3, we present the results on FMoW and ChEMBL-STRING of introducing consistency regularization in the setting where only fixed relations are

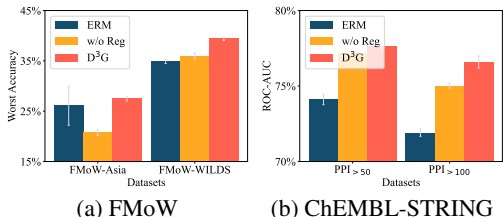

| (a) FMoW | (b) ChEMBL-STRING |
|---|---|

Figure 3: Performance comparison w.r.t. consistency regularization. Only fixed relations are used.

used. According to the results, we observe better performance when introducing consistency regularization, indicating its effectiveness in learning domain-specific models by strengthening the correlations between the domain-specific functions.

**How do domain relations benefit performance?** Our theoretical analysis shows that utilizing appropriate domain relations can enhance performance compared to simply averaging predictions from all domain-specific functions. To test this, we conducted analysis in FMoW and ChEMBL-STRING, comparing the following variants of relations: (1) no relations used; (2) fixed relations only;

Table 1: Performance comparison between D$^3$G and other baselines. Here, all baselines use domain meta-data as features. The discrepancy in performance between our results and those reported on the leaderboard for FMoW-WILDS because we incorporate domain meta-data as features for all baselines. We **bold** the best results and underline the second best results.

| | TPT-48 (MSE ↓) | | FMoW (Worst Acc. ↑) | | ChEMBL-STRING (ROC-AUC ↑) | |
| | N (24) → S (24) | E (24) → W (24) | FMoW-Asia | FMoW-WILDS | PPI$_{>50}$ | PPI$_{>100}$ |
| | Region Shift | Region Shift | Region Shift | Region-Time Shift | Protein Shift | Protein Shift |
|---|---|---|---|---|---|---|
| ERM | $0.445 \pm 0.029$ | $0.328 \pm 0.033$ | $26.05 \pm 3.84\%$ | $34.87 \pm 0.41\%$ | $74.11 \pm 0.35\%$ | $71.91 \pm 0.24\%$ |
| GroupDRO | $0.413 \pm 0.045$ | $0.434 \pm 0.082$ | $26.24 \pm 1.85\%$ | $31.16 \pm 2.12\%$ | $73.98 \pm 0.25\%$ | $71.55 \pm 0.59\%$ |
| IRM | $0.429 \pm 0.043$ | $\underline{0.262 \pm 0.034}$ | $25.02 \pm 2.38\%$ | $32.54 \pm 1.92\%$ | $52.71 \pm 0.50\%$ | $51.73 \pm 1.54\%$ |
| IB-IRM | $0.416 \pm 0.009$ | $\underline{0.272 \pm 0.026}$ | $26.30 \pm 1.51\%$ | $34.94 \pm 1.38\%$ | $52.12 \pm 0.91\%$ | $52.33 \pm 1.06\%$ |
| IB-ERM | $0.458 \pm 0.032$ | $0.273 \pm 0.030$ | $26.78 \pm 1.34\%$ | $35.52 \pm 0.79\%$ | $74.69 \pm 0.14\%$ | $\underline{73.32 \pm 0.21\%}$ |
| V-REx | $0.412 \pm 0.042$ | $0.343 \pm 0.021$ | $26.63 \pm 0.93\%$ | $\underline{37.64 \pm 0.92\%}$ | $71.46 \pm 1.47\%$ | $69.37 \pm 0.85\%$ |
| DANN | $0.394 \pm 0.019$ | $0.515 \pm 0.156$ | $25.62 \pm 1.59\%$ | $33.78 \pm 1.55\%$ | $73.49 \pm 0.45\%$ | $72.22 \pm 0.10\%$ |
| CORAL | $0.401 \pm 0.022$ | $0.283 \pm 0.048$ | $25.87 \pm 1.97\%$ | $36.53 \pm 0.15\%$ | $\underline{75.42 \pm 0.15\%}$ | $73.10 \pm 0.14\%$ |
| MMD | $0.409 \pm 0.067$ | $0.279 \pm 0.026$ | $25.06 \pm 2.19\%$ | $35.48 \pm 1.81\%$ | $75.11 \pm 0.27\%$ | $73.30 \pm 0.50\%$ |
| RSC | $0.421 \pm 0.040$ | $0.330 \pm 0.068$ | $25.73 \pm 0.70\%$ | $34.59 \pm 0.42\%$ | $74.83 \pm 0.68\%$ | $72.47 \pm 0.38\%$ |
| CAD | n/a | n/a | $26.13 \pm 1.82\%$ | $35.17 \pm 1.73\%$ | $75.17 \pm 0.64\%$ | $72.92 \pm 0.39\%$ |
| SelfReg | n/a | n/a | $24.81 \pm 1.77\%$ | $37.33 \pm 0.87\%$ | $\underline{75.42 \pm 0.42\%}$ | $72.63 \pm 0.71\%$ |
| Mixup | $0.574 \pm 0.030$ | $0.357 \pm 0.011$ | $\underline{26.99 \pm 1.27\%}$ | $35.67 \pm 0.53\%$ | $74.40 \pm 0.54\%$ | $71.31 \pm 1.06\%$ |
| LISA | $0.467 \pm 0.032$ | $0.345 \pm 0.014$ | $26.05 \pm 2.09\%$ | $34.59 \pm 1.28\%$ | $74.30 \pm 0.59\%$ | $71.45 \pm 0.44\%$ |
| MAT | $0.423 \pm 0.027$ | $0.291 \pm 0.024$ | $25.92 \pm 2.83\%$ | $35.07 \pm 0.84\%$ | $74.73 \pm 0.30\%$ | $72.07 \pm 0.81\%$ |
| AdaGraph | n/a | n/a | $25.91 \pm 0.59\%$ | $35.42 \pm 0.55\%$ | $74.02 \pm 0.42\%$ | $72.10 \pm 0.06\%$ |
| RaMoE | $0.372 \pm 0.035$ | $0.311 \pm 0.060$ | $26.65 \pm 0.46\%$ | $36.51 \pm 0.71\%$ | $74.99 \pm 0.22\%$ | $71.48 \pm 0.49\%$ |
| mDSDI | $0.445 \pm 0.027$ | $0.315 \pm 0.089$ | $25.54 \pm 0.46\%$ | $36.35 \pm 0.45\%$ | $75.09 \pm 0.47\%$ | $71.23 \pm 0.69\%$ |
| ADDAR | $0.403 \pm 0.061$ | $0.287 \pm 0.040$ | $25.87 \pm 1.01\%$ | $35.77 \pm 0.70\%$ | $74.55 \pm 0.54\%$ | $71.93 \pm 0.33\%$ |
| GRDA | $0.373 \pm 0.040$ | $0.355 \pm 0.068$ | $26.57 \pm 0.70\%$ | $34.41 \pm 0.42\%$ | $75.01 \pm 0.68\%$ | $73.57 \pm 0.38\%$ |
| DRM | $0.571 \pm 0.038$ | $0.557 \pm 0.027$ | $25.22 \pm 2.33\%$ | $36.39 \pm 0.76\%$ | $74.34 \pm 0.48\%$ | $72.41 \pm 0.76\%$ |
| LLE | $0.603 \pm 0.041$ | $0.467 \pm 0.047$ | $26.37 \pm 1.19\%$ | $35.83 \pm 1.00\%$ | $74.01 \pm 0.63\%$ | $71.68 \pm 0.61\%$ |
| DDN | $0.537 \pm 0.024$ | $0.601 \pm 0.038$ | $26.77 \pm 1.72\%$ | $35.13 \pm 0.62\%$ | $75.17 \pm 0.61\%$ | $72.71 \pm 0.59\%$ |
| TRO | $\underline{0.371 \pm 0.054}$ | $0.281 \pm 0.066$ | $26.87 \pm 1.26\%$ | $37.48 \pm 0.55\%$ | $74.85 \pm 0.27\%$ | $72.49 \pm 0.36\%$ |
| **D$^3$G (ours)** | $\mathbf{0.342 \pm 0.019}$ | $\mathbf{0.236 \pm 0.063}$ | $\mathbf{28.12 \pm 0.28\%}$ | $\mathbf{39.47 \pm 0.57\%}$ | $\mathbf{78.67 \pm 0.16\%}$ | $\mathbf{77.24 \pm 0.30\%}$ |

(3) learned relations only; and (4) both fixed and learned relations. Our results, presented in Table 3 (Full results: Appendix H.1), first indicate that using fixed relations outperforms averaging predictions, which confirms our theoretical findings and highlights the importance of using appropriate relations. However, only using learned relations resulted in a performance that is worse than using no relations at all, indicating that it is challenging to learn relations without any informative signals (e.g., fixed relations). Finally, combining learned and fixed relations results in the best performance, highlighting the importance of using learned relations to find more accurate relations for each problem.

## 5.4 COMPARISON OF D$^3$G WITH DOMAIN-SPECIFIC FINE-TUNING

As stated in the introduction, a simple way to create a domain-specific model is by fine-tuning a generic model trained by empirical risk minimization (ERM) on reweighted training data using domain relations. In this section, we compare our proposed model D$^3$G with this approach (referred to as RW-FT) and present the results in Table 2. We also include the performance of the strongest baseline (CORAL) for comparison. The results show that RW-FT outperforms ERM and CORAL, further confirming the effectiveness of using domain distances to improve out-of-distribution generalization. Additionally, D$^3$G performs better than RW-FT. This may be due to the fact that using separate models for each training domain allows for more effective capture of domain-specific information.

## 5.5 ANALYSIS OF RELATION REFINEMENT

In this section, we conduct a qualitative analysis to determine if the relations learned can reflect application-specific information and improve the fixed relations extracted from domain meta-data. Specifically, we select three countries - Turkey, Syria, and Saudi Arabia from FMoW-Asia and visualize the fixed relations and learned relations among them in Figure 4. Additionally, we visualize one multi-unit residential area from each of the three countries. We observe that although Turkey is geographically close to other two countries in the Middle East (as shown by the fixed relations), its architecture style is influenced by Europe. Therefore, the learned relations refine the fixed relations and weaken the distances between Turkey and Saudi Arabia and Syria.

Table 2: Comparison between D$^3$G with domain-specific fine-tuning. Full results: Appendix H.2.

| Model | FMoW (Worst Acc. ↑) | | ChEMBL (AUC ↑) | |
|---|---|---|---|---|
| | Asia | WILDS | PPI$_{>50}$ | PPI$_{>100}$ |
| ERM | 26.05% | 34.87% | 74.11% | 71.91% |
| CORAL | 25.87% | 36.53% | 75.42% | 73.10% |
| RW-FT | 27.03% | 36.39% | 76.31% | 74.30% |
| **D$^3$G** | **28.12%** | **39.47%** | **78.67%** | **77.24%** |

Table 3: Comparison of using different relations. The results on FMoW and ChEMBL-STRING are reported. When no relations are used, we take the average of predictions across all domains.

| Fixed relations | Learned relations | FMoW (Worst Acc. ↑) | | ChEMBL (AUC ↑) | |
|---|---|---|---|---|---|
| | | Asia | WILDS | PPI$_{>50}$ | PPI$_{>100}$ |
| | | 26.93% | 35.32% | 76.17% | 73.38% |
| ✓ | | 27.43% | 39.37% | 77.66% | 76.59% |
| | ✓ | 21.18% | 36.41% | 77.09% | 75.57% |
| ✓ | ✓ | **28.12%** | **39.47%** | **78.67%** | **77.24%** |

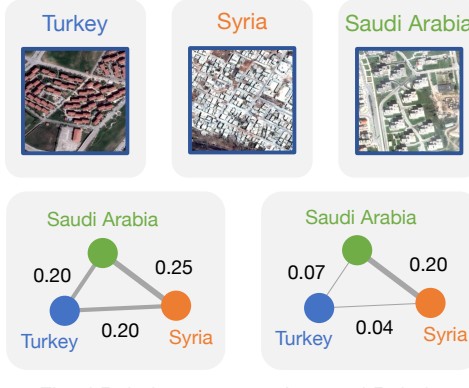

Figure 4: Analysis of relation learning. Top figures show the multi-unit residential areas of Turkey, Syria, and Saudi Arabia. Bottom figures illustrate both fixed relations and learned relations.

# 6 RELATED WORK

In this section, we discuss the related work from the following two categories: out-of-distribution generalization, ensemble learning, and test-time adaptation (Appendix B).

**Out-of-distribution Generalization.** To improve out-of-distribution generalization, the first line of works aligns representations across domains to learn invariant representations by (1) minimizing the divergence of feature distributions (Long et al., 2015; Tzeng et al., 2014; Ganin et al., 2016a; Li et al., 2018b); (2) generating more domains and enhancing the consistency among representations (Shu et al., 2021; Wang et al., 2020; Xu et al., 2020; Yan et al., 2020; Yue et al., 2019; Zhou et al., 2020). Another line of works aims to find a predictor that is invariant across domains by imposing an explicit regularizer (Arjovsky et al., 2019; Ahuja et al., 2021a; Guo et al., 2021; Khezeli et al., 2021; Koyama and Yamaguchi, 2020; Krueger et al., 2021b; Koyama and Yamaguchi, 2020) or selectively augmenting more examples (Yao et al., 2022b;a; Gao et al.). Recent studies explored the concept of learning domain-specific models (Li et al., 2022; Pagliardini et al., 2022; Zhang et al., 2023; 2022b). However, in comparison to these approaches, D$^3$G stands out by leveraging domain meta-data, incorporating consistency regularization and domain relation refinement techniques, to more effectively capture domain relations. Notably, even when compared to these prior domain-specific learning approaches that incorporate meta-data, D$^3$G consistently achieves superior performance, underscoring its effectiveness in domain relation modeling.

**Ensemble methods.** Our approach is closely related to ensemble methods, such as those that aggregate the predictions of multiple learners (Hansen and Salamon, 1990; Dietterich, 2000; Lakshminarayanan et al., 2017) or selectively combine the prediction from multiple experts (Jordan and Jacobs, 1994; Eigen et al., 2013; Shazeer et al., 2017; Dauphin et al., 2017). When distribution shift occurs, prior works have attempted to solve the underspecification problem by learning a diverse set of functions with the help of unlabeled data (Teney et al., 2021; Pagliardini et al., 2022; Lee et al., 2022). These methods aim to resolve the underspecification problem in the training data and disambiguate the model, thereby improving out-of-distribution robustness. Unlike prior works that rely on ensemble models to address the underspecification problem and improve out-of-distribution robustness, our proposed D$^3$G takes a conceptually different approach by constructing domain-specific models.

# 7 CONCLUSION

In summary, the paper presents a novel method called D$^3$G for tackling the issue of domain shifts in real-world machine learning scenarios. The approach leverages the connections between different domains to enhance the model's robustness and employs a domain-relationship aware weighting system for each test domain. We evaluate the effectiveness of D$^3$G on various datasets and observe that it consistently surpasses current methods, resulting in substantial performance enhancements.

## REPRODUCIBILITY STATEMENT

To make sure our work can be easily reproduced, we've outlined the training and test steps in Algorithm 1. Our theory in Section 4 is supported by proofs in the Appendix A. We've also provided more information about our experiments and settings in Appendix D, along with a description of the dataset in Appendix E. Code to reproduce our results will be made publicly available.

## ACKNOWLEDGEMENT

We thank Linjun Zhang, Hao Wang, and members of the IRIS lab for the many insightful discussions and helpful feedback. This research was supported by Apple and Juniper Networks. CF is a CIFAR fellow.

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

# A    DETAILED PROOFS

## A.1    PROOF OF THEOREM 4.1

To prove theorem 4.1, let us first define an intermediate function

$$h_{im}^{(t)} = \frac{\sum_{i=1}^{N^{tr}} a_{it} h^{(i)}}{\sum_{k=1}^{N^{tr}} a_{kt}} \tag{10}$$

We then define the event $E_n = \{\sum_{i=1}^{N^{tr}} a_{it} > 0\}$. Based on our assumption $\mathbb{E}[(\hat{h}^{(d)}(e(x)) - h^{(d)}(e(x)))^2] = O(\frac{C(\mathcal{H})}{n_d})$ and $n_d \gtrsim n$ for each domain $d$. On the event $E_n$, we have that

$$
\begin{aligned}
\mathbb{E}[(h_{im}^{(t)}(e(x)) - \hat{h}^{(t)}(e(x)))^2] &\leq \frac{\sum_{i=1}^{N^{tr}} a_{it} \cdot \mathbb{E}\left[(\hat{h}^{(i)}(e(x)) - h^{(i)}(e(x)))^2\right]}{(\sum_{k=1}^{N^{tr}} a_{kt})^2} \\
&\leq \frac{\max_i \mathbb{E}\left[(\hat{h}^{(i)}(e(x)) - h^{(i)}(e(x)))^2\right]}{\sum_{k=1}^{N^{tr}} a_{kt}} \\
&= O\left(\frac{C(\mathcal{H})}{n \sum_{k=1}^{N^{tr}} a_{kt}}\right),
\end{aligned}
\tag{11}
$$

Moreover, since $\|h^{(i)} - h^{(j)}\|_\infty \leq G \cdot \|Z^{(i)} - Z^{(j)}\| \leq G \cdot B$ when $\|Z^{(i)} - Z^{(j)}\| \leq B$, we have that on the scenario $E_n$,

$$
\left| h_{im}^{(t)} - h^{(t)} \right| = \left| \frac{\sum_{i=1}^{N^{tr}} a_{it}(h^{(i)} - h^{(t)})}{\sum_{k=1}^{N^{tr}} a_{kt}} \right| = \left| \frac{\sum_{i=1}^{N^{tr}} \mathbf{1}\{\|Z^{(i)} - Z^{(t)}\| < B\}(h^{(i)} - h^{(t)})}{\sum_{k=1}^{N^{tr}} \mathbf{1}\{\|Z^{(k)} - Z^{(t)}\| < B\}} \right| \leq G \cdot B.
\tag{12}
$$

On the other hand, on the complement event $E_n^c$, as the denominator equals to 0 by our definition, we have $h_{im}^{(t)}(e(x)) = 0$ and therefore

$$\left| h_{im}^{(t)}(e(x)) - h^{(t)}(e(x)) \right|^2 = (h^{(t)})^2(e(x)). \tag{13}$$

Consequently, we have

$$\left| h_{im}^{(t)}(e(x)) - h^{(t)}(e(x)) \right|^2 \leq G^2 B^2 + (h^{(t)})^2(e(x)) \cdot \mathbf{1}_{E_n^c}. \tag{14}$$

Therefore,

$$\mathbb{E}\left[(\hat{h}^{(t)} - h^{(t)})^2\right] \lesssim \mathbb{E}\left[\frac{C(\mathcal{H})}{n \sum_{k=1}^{N^{tr}} a_{kt}} \cdot \mathbf{1}_{E_n}\right] + B^2 + \mathbb{E}\left[(h^{(t)})^2(e(x)) \cdot \mathbf{1}_{E_n^c}\right].$$

To bound the first term, we let $S = \sum_{i=1}^{N^{tr}} \mathbf{1}\{\|Z^{(t)} - Z^{(i)}\| < B\}$. Since $Z^{(d)}$ are uniformly distributed on $[0,1]^r$, we have that $S \sim Binomial(N^{tr}, q)$ with $q = \mathbb{P}(\|Z - Z^{(t)}\| < B)$. Using the property of binomial distribution, we have

$$\mathbb{E}\left[\frac{\mathbf{1}\{S > 0\}}{S}\right] \lesssim \frac{1}{N^{tr} q} \lesssim \frac{1}{N^{tr} B^r}. \tag{15}$$

Therefore the first term

$$\mathbb{E}\left[\frac{C(\mathcal{H})}{n \sum_{k=1}^{N^{tr}} a_{kt}} \cdot \mathbf{1}_{E_n}\right] \lesssim \frac{C(\mathcal{H})}{n N^{tr} B^r}. \tag{16}$$

The third term can be bounded as

$$\mathbb{E}\left[(h^{(t)})^2(e(x)) \cdot \mathbf{1}_{E_n^c}\right] \le \sup(h^{(t)})^2(e(x))\mathbb{E}[(1-q)^{N^{tr}}] \lesssim \sup(h^{(t)})^2(e(x))\frac{1}{N^{tr}q} \lesssim \frac{1}{N^{tr}B^r}. \tag{17}$$

Combining all the pieces, we get

$$\mathbb{E}\left[(\hat{h}^{(t)} - h^{(t)})^2\right] \lesssim B^2 + \frac{C(\mathcal{H})/n}{N^{tr}B^r}. \tag{18}$$

Therefore, when $l$ is Lipshictz with respect to the first argument, we have that

$$\mathbb{E}_{(x,y)\sim P_t}[\ell(\hat{f}^{(t)}(x), y)] - \mathbb{E}_{(x,y)\sim P_t}[\ell(f^{(t)}(x), y)] \le \mathbb{E}\left[|\hat{h}^{(t)} - h^{(t)}|\right]$$

$$\le \sqrt{\mathbb{E}[(\hat{h}^{(t)} - h^{(t)})^2]} \lesssim B + \sqrt{\frac{C(\mathcal{H})/n}{N^{tr}B^r}}. \tag{19}$$

## A.2 PROOF OF PROPOSITION 4.2

If we treat all training domains are equally important, i.e., $a_{it} = 1$ for all $i$ and $t$, we have that

$$\hat{h}^{(t)} = \frac{\sum_{i=1}^{N^{tr}} a_{it}\hat{h}^{(i)}}{\sum_{k=1}^{N^{tr}} a_{kt}} = \frac{1}{N^{tr}}\sum_{i=1}^{N^{tr}} \hat{h}^{(i)}. \tag{20}$$

As it is simply the average of predictive values, we denote it by $\hat{h}^{(avg)}$.

In order to show that such an estimator performs worse than $\hat{h}$ in the minimax sense, all we need is to find an $h \in \mathcal{H}$ so that $R_h(h^{(avg)}(f(x))) = \Omega(1)$ even for $N, n \to \infty$.

We simply let $d \sim U[0,1]$, $e(x) \in \mathcal{N}(0,1)$ and $h^{(d)}(e(x)) = d \cdot e(x)$. Under such a setting, we have that

$$\hat{h}^{(avg)} = \frac{1}{2}e(x),$$

and therefore

$$\mathbb{E}[\hat{h}^{(avg)}(e(x)) - h^{(d)}(e(x))^2] = \mathbb{E}\left[(d - \frac{1}{2})^2 e^2(x)\right] = \frac{1}{12} = \Omega(1). \tag{21}$$

We complete the proof.

## B ADDITIONAL RELATED WORK: DISCUSSION WITH TEST-TIME ADAPTATION

Traditional test-time adaptation approaches for domain generalization typically involve utilizing unlabeled target data to adapt the model. These approaches can be categorized into two main groups: (1) jointly optimizing the model with supervised or self-supervised loss during test time (Bartler et al., 2022; Liu et al., 2021; Sun et al., 2020); (2) adapting the model solely during the test phase, which includes techniques such as adapting batch normalization (Lim et al., 2023; Schneider et al., 2020; Zhao et al., 2023), maximizing prediction consistency (Wang et al., 2022a; Zhang et al., 2022a), etc. In contrast to these test-time adaptation methods, D³G takes a different approach by not relying on unlabeled data from the test domain during inference. Instead, D3G leverages domain meta-data, which is distinct from unlabeled data. This fundamental distinction makes D3G more suitable for domain generalization, as it allows the model to generalize to unseen domains without accessing their data during training.

## C DETAILED DESCRIPTION OF BASELINES

In this work, we compare D³G to a large number of algorithms that span different learning strategies. We group them according to their categories, and provide detailed descriptions for each algorithm below.

- *vanilla*: ERM (Vapnik, 1999) minimizes the average empirical loss across all training data.

- *distributionally robust optimization*: GroupDRO (Sagawa et al., 2020) optimizes the worst-domain loss.

- *invariant learning*: IRM (Arjovsky et al., 2019) learns invariant predictors that perform well across different domains. IB-IRM and IB-ERM (Ahuja et al., 2021b) performs IRM and ERM respectively with the information bottleneck constraint. V-REx (Krueger et al., 2021a) proposes a penalty on the variance of training risks. DANN (Ganin et al., 2016b) employs an adversarial network to match feature distributions. CORAL (Sun and Saenko, 2016) matches the mean and covariance of feature distributions. MMD (Li et al., 2018a) matches the maximum mean discrepancy (Gretton et al., 2012) of feature distributions. RSC (Huang et al., 2020) discards the representations associated with the higher gradients in each epoch, forcing the model to predict with the remaining information. CAD (Ruan et al., 2022) introduces a contrastive adversarial domain bottleneck to enforce support match using a KL divergence. SelfReg (Kim et al., 2021) utilizes the self-supervised contrastive losses to learn domain-invariant representation. Mixup (Xu et al., 2020) performs ERM on linear interpolations of examples from random pairs of domains and their labels. LISA (Yao et al., 2022b) builds upon Mixup but interpolates samples with the same label but different domains, MAT (Wang et al., 2022b) conducts adversarial training with combinations of domain-wise multiple perturbations.

- *domain-specific learning*: AdaGraph (Mancini et al., 2019) introduces GraphBN to provide each domain with its own BN statistics. RaMoE (Dai et al., 2021) use an effective voting-based mixture mechanism to dynamically leverage cdomain-specific characteristics. mDSDI (Bui et al., 2021) introduce a meta-optimization training framework to boost domain-specific learning. AF-FAR (Qin et al., 2022) learns domain-specific and domain-invariant representations, and fuse them dynamically. GRDA (Xu et al., 2022) generalize the adversarial learning framework with a graph discriminator encoding domain adjacencys.. DRM (Zhang et al., 2022b) utilizes test-time model selection and adaption to ensemble domain-specific classifiers. LLE (Li et al., 2023) aggregates the predictions from the ensemble of the last layers based on the distributional shift classifier. DDN (Zhang et al., 2023) performs domain-based constrastive learning to decouple both domain invariant and task-specific domain variations. TRO (Qiao and Peng, 2023) integrates distributional topology in a principled optimization framework.

## D   DETAILED EXPERIMENTAL SETUPS AND HYPERPARAMETERS

In this section, we detail our model selection for all datasets, where we use the same model architectures and use the same input $(x, y, d)$ for all approaches for fair comparison, where domain meta-data is used as features. Following (Xu et al., 2022), we adopt a two layer MLP network, and use no data augmentation for DG-15 and TPT-48. For the FMoW dataset, we fix the network architecture as the pretrained DenseNet-121 model (Huang et al., 2017) and use the same data augmentation protocol as (Koh et al., 2021b): random crop and resize to 224 × 224 pixels, and normalization using the ImageNet channel statistics. For the ChEMBL-STRING dataset, we use graph isomorphism network (GIN) (Xu et al., 2019) as the backbone network for all algorithms, and use no data augmentation. We use an additional two layer MLP network to incorperate the domain meta-data as features for all approaches.

We list the hyperparameters in Table 4 for all datasets.

Table 4: Hyperparameters for D³G on all datasets.

| Hyperparameters | DG-15 | TPT-48 | FMoW | ChEMBL-STRING |
|---|---|---|---|---|
| Learning Rate | 1e-5 | 2e-3 | 1e-4 | 1e-4 |
| Weight Decay | 5e-4 | 5e-4 | 0 | 0 |
| Batch Size | 10 | 64 | 32 | 30 |
| Epochs | 30 | 40 | 60 | 100 |
| Loss Balanced Coefficient $\lambda$ | 0.5 | 0.5 | 0.5 | 0.5 |
| Relation Combining Coefficient $\beta$ | 0.8 | 0.5 | 0.8 | 0.5 |

# E  ADDITIONAL DESCRIPTION OF DATASETS

## E.1  TPT-48

TPT-48 is a real-world weather prediction dataset from the nClimDiv and nClimGrid (Vose et al., 2014) databases, which contains the monthly average temperature for the 48 contiguous states in the US from 2008 to 2019. The task is to forecast the next 6 months' temperature based on the previous 6 months' temperature. Each state is treated as a domain, and the domain meta-data is defined as the geographical location of each state. Following Xu et al. (2022), we consider two dataset splits: I. N (24) → S (24): generalizing from the 24 states in the north to the 24 states in the south; II. E (24) → W (24): generalizing from the 24 states in the east to the 24 states in the west. A 0/1 adjacency matrix is used as the fixed domain similarity matrix, where $a_{ij}^g = 1$ represents that states $i$ and $j$ are geographically connected. We show the fixed relations for N (24) → S (24) and E (24) → W (24) in TPT-48. For the 24 test domains in these two tasks, we further random split them into 12 validation domains and 12 test domains.

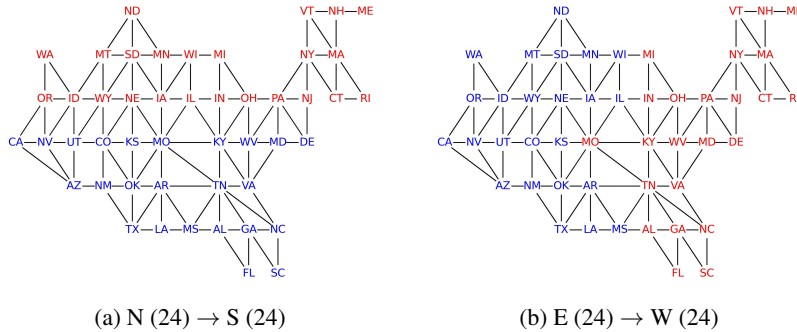

(a) N (24) → S (24)                    (b) E (24) → W (24)

Figure 5: Fixed relation graphs for the two tasks on TPT-48, with red nodes indicating training domains and blue nodes indicating test domains. Left: Generalization from the 24 states in the north to the 24 states in the south. Right: Generalization from the 24 states in the east to the 24 states in the west.

## E.2  FMoW

Similar to TPT-48, we use a 0/1 adjacency matrix to formulate the fixed domain relations in FMoW, where $a_{ij}^g = 1$ represents that regions $i$ and $j$ are geographically connected. For the construction of FMoW-Asia, we choose 18 countries or special administrative regions in Asia with most satellite images, and regions belong to the same sub-continent (Eastern Asia, Western Asia, Central Asia, Southern Asia and South-eastern Asia) are connected. The number of training, validation and test domains are 8, 5, 5 respectively.

## E.3  CHEMBL-STRING

We follow the dataset proposed in SGNN-EBM (Liu et al., 2022), ChEMBL-STRING. It is multi-domain dataset with explicit domain relation. The three key steps are as follows:

- Filtering molecules. Among 456,331 molecules in the LSC dataset, 969 are filtered out following the pipeline in (Hu et al., 2019).
- Querying the PPI scores. Then we obtain the PPI scores by quering the ChEMBL (Mendez et al., 2018) and STRING (Szklarczyk et al., 2019) databases.
- Finally, we calculate the edge weights $w_{ij}$, *i.e.*, domain relation score, for domain $i$ and $j$ in the domain relation graph to be $\max\{\text{PPI}(m_i, m_j) : m_i \in \mathcal{M}_i, m_i \in \mathcal{M}_j\}$, where $\mathcal{M}_i$ denotes the protein set of domain $i$. The resulting domain relation graph has 1,310 nodes and 9,172 edges with non-zero weights.

The statistics of the resulting ChEMBL-STRING dataset with two thresholds can be found at Table 5.

Table 5: Statistics about PPI$_{>50}$ and PPI$_{>100}$ subsets in ChEMBL-STRING, where we use proteins as domains. Sparsity here is defined as the ratio of zero values in the domain relation graph.

| Threshold | # Samples | # Proteins | Sparsity | # Train Proteins | # Valid Proteins | # Test Proteins |
|---|---|---|---|---|---|---|
| PPI$_{>50}$ | 87908 | 140 | 0.914 | 92 | 19 | 29 |
| PPI$_{>100}$ | 58823 | 121 | 0.911 | 73 | 24 | 24 |

# F  ADDITIONAL RESULTS OF ILLUSTRATIVE TOY TASK

The full results on DG-15 are reported in Table 6.

Table 6: Full Results of domain shifts on DG-15. The standard deviation is computed across three seeds.

| Model | ERM | GroupDRO | IRM | IB-IRM | IB-ERM | V-REx |
|---|---|---|---|---|---|---|
| Accuracy | $44.0 \pm 4.6\%$ | $47.7 \pm 9.0\%$ | $43.9 \pm 5.1\%$ | $45.4 \pm 5.7\%$ | $43.1 \pm 1.9\%$ | $44.0 \pm 4.1\%$ |

| Model | DANN | CORAL | MMD | RSC | CAD | SelfReg |
|---|---|---|---|---|---|---|
| Accuracy | $43.1 \pm 4.5\%$ | $43.5 \pm 1.5\%$ | $41.3 \pm 1.0\%$ | $58.2 \pm 4.3\%$ | $43.3 \pm 2.8\%$ | $40.7 \pm 0.7\%$ |

| Model | Mixup | LISA | MAT | RaMoE | mDSDI | AFFAR |
|---|---|---|---|---|---|---|
| Accuracy | $41.3 \pm 3.9\%$ | $47.4 \pm 0.9\%$ | $39.6 \pm 1.4\%$ | $53.7 \pm 1.9\%$ | $45.2 \pm 1.0\%$ | $58.2 \pm 4.3\%$ |

| Model | GRDA | DRM | LLE | DDN | TRO | D$^3$G (ours) |
|---|---|---|---|---|---|---|
| Accuracy | $59.5 \pm 4.1\%$ | $61.4 \pm 2.3\%$ | $58.8 \pm 3.1\%$ | $\underline{66.2 \pm 2.0\%}$ | $56.3 \pm 2.7\%$ | $\mathbf{77.5 \pm 2.5\%}$ |

# G  ADDITIONAL RESULTS OF REAL WORLD DOMAIN SHIFTS

## G.1  FULL RESULTS ON FMOW

The full results on FMoW are reported in Table 7.

## G.2  FULL RESULTS ON CHEMBL-STRING

The full results on ChEMBL-STRING are reported in Table 8.

## G.3  INCORPORATING DOMAIN RELATIONS INTO THE DOMAIN-SPECIFIC LEARNING

To better emphasize the effectiveness of our domain relation refinement and consistency regularization, we present additional evidence in Table 9 by incorporating existing domain relations into previous domain-specific learning methods. It is important to note that we also include domain meta-data in these models. As shown in Table 9, the introduction of existing domain relations does improve the performance of these domain-specific learning approaches. However, even with the incorporation of these relations, D$^3$G outperforms them significantly. This outcome is not surprising, considering that refining domain relations with consistency regularization enables us to better capture domain similarity, leading to superior performance.

# H  ADDITIONAL RESULTS OF ANALYSIS

## H.1  FULL RESULTS OF THE ANALYSIS WITH DIFFERENT RELATIONS

In Table 10, we report the full results of the analysis of different types of relations.

Table 7: Full results of domain shifts on FMoW. The standard deviation is computed across three seeds.

| | FMoW-Aisa | | FMoW-WILDS | |
|---|---|---|---|---|
| | Worst Acc. | Average Acc. | Worst Acc. | Average Acc. |
| ERM | $26.05 \pm 3.84\%$ | $35.50 \pm 0.20\%$ | $34.87 \pm 0.41\%$ | $52.64 \pm 0.30\%$ |
| GroupDRO | $26.24 \pm 1.85\%$ | $34.14 \pm 0.15\%$ | $31.16 \pm 2.12\%$ | $46.79 \pm 0.08\%$ |
| IRM | $25.02 \pm 2.38\%$ | $33.28 \pm 0.68\%$ | $32.54 \pm 1.92\%$ | $50.39 \pm 0.66\%$ |
| IB-IRM | $26.30 \pm 1.51\%$ | $35.43 \pm 0.37\%$ | $34.94 \pm 1.38\%$ | $51.90 \pm 0.27\%$ |
| IB-ERM | $26.78 \pm 1.34\%$ | $35.65 \pm 0.62\%$ | $35.52 \pm 0.79\%$ | $52.36 \pm 0.31\%$ |
| V-REx | $26.63 \pm 0.93\%$ | $35.71 \pm 0.21\%$ | $\underline{37.64 \pm 0.92\%}$ | $52.89 \pm 0.10\%$ |
| DANN | $25.62 \pm 1.59\%$ | $34.53 \pm 0.76\%$ | $\overline{33.78 \pm 1.55\%}$ | $50.50 \pm 0.25\%$ |
| CORAL | $25.87 \pm 1.97\%$ | $35.43 \pm 0.12\%$ | $36.53 \pm 0.15\%$ | $51.89 \pm 0.35\%$ |
| MMD | $25.06 \pm 2.19\%$ | $33.77 \pm 0.57\%$ | $35.48 \pm 1.81\%$ | $50.17 \pm 0.17\%$ |
| RSC | $25.73 \pm 0.70\%$ | $34.27 \pm 0.38\%$ | $34.59 \pm 0.42\%$ | $51.32 \pm 0.77\%$ |
| CAD | $26.13 \pm 1.82\%$ | $34.97 \pm 0.41\%$ | $35.17 \pm 1.73\%$ | $50.92 \pm 0.35\%$ |
| SelfReg | $24.81 \pm 1.77\%$ | $36.27 \pm 0.50\%$ | $37.33 \pm 0.87\%$ | $51.71 \pm 0.28\%$ |
| Mixup | $26.99 \pm 1.27\%$ | $36.41 \pm 0.31\%$ | $35.67 \pm 0.53\%$ | $\underline{53.50 \pm 0.11\%}$ |
| LISA | $\overline{26.05 \pm 2.09\%}$ | $\overline{35.17 \pm 0.69\%}$ | $34.59 \pm 1.28\%$ | $\overline{50.95 \pm 0.13\%}$ |
| MAT | $25.92 \pm 2.83\%$ | $34.68 \pm 0.23\%$ | $35.07 \pm 0.84\%$ | $52.11 \pm 0.25\%$ |
| AdaGraph | $25.91 \pm 0.59\%$ | $34.18 \pm 0.17\%$ | $35.42 \pm 0.55\%$ | $50.74 \pm 0.62\%$ |
| RaMoE | $26.65 \pm 0.46\%$ | $34.37 \pm 0.36\%$ | $36.51 \pm 0.71\%$ | $52.29 \pm 0.30\%$ |
| mDSDI | $25.54 \pm 0.46\%$ | $34.63 \pm 0.35\%$ | $36.35 \pm 0.45\%$ | $52.14 \pm 0.49\%$ |
| ADDAR | $25.87 \pm 1.01\%$ | $34.04 \pm 0.83\%$ | $35.77 \pm 0.70\%$ | $52.23 \pm 0.50\%$ |
| GRDA | $26.57 \pm 0.70\%$ | $34.47 \pm 0.62\%$ | $34.41 \pm 0.42\%$ | $53.03 \pm 0.97\%$ |
| DRM | $26.37 \pm 1.19\%$ | $35.22 \pm 0.47\%$ | $35.83 \pm 1.00\%$ | $52.66 \pm 0.41\%$ |
| LLE | $25.22 \pm 2.33\%$ | $35.67 \pm 0.19\%$ | $36.39 \pm 0.76\%$ | $51.80 \pm 0.19\%$ |
| DDN | $26.77 \pm 1.72\%$ | $35.30 \pm 0.31\%$ | $35.13 \pm 0.62\%$ | $52.36 \pm 0.34\%$ |
| TRO | $26.87 \pm 1.26\%$ | $35.03 \pm 0.42\%$ | $37.48 \pm 0.55\%$ | $52.66 \pm 0.28\%$ |
| **D³G (ours)** | **$28.12 \pm 0.28\%$** | **$37.71 \pm 0.48\%$** | **$39.47 \pm 0.57\%$** | **$54.36 \pm 0.12\%$** |

## H.2 Full Results of Comparison with Domain-specific Fine-tuning

In Table 11, we report the full results of comparison with domain-specific fine-tuning.

## H.3 Full Results of Comparison with using separate feature extractors for each training domain.

We conduct an additional comparison between $D^3G$ and one variant, which employs separate feature extractors for each training domain. The results are reported in Table 12. According to the results, $D^3G$ shows superior performance compared to using separate feature extractors for each training domain. This underscores the importance of employing a shared feature extractor to learn a universal representation, while domain-specific heads can further identify domain-specific features. Additionally, our $D^3G$ model demonstrates superior efficiency compared to this variant since it utilizes a shared feature extractors while the variant needs $N^{tr}$ feature extractors.

## H.4 Full Results of Comparison with its variant that use limited domain information

We further compare $D^3G$ with two addition variants, in case where we only have limited domain information. The two variants with the corresponding analysis are described as:

**$D^3G$ w/o domain split information**: In situations where domain split information is unavailable, we employ a clustering approach to group all training data into several clusters, with the number of clusters matching the actual number of domains. Each cluster is treated as a separate domain, and domain relations are established based on the distances between the clustering centers.

Table 8: Full results of domain shifts on ChEMBL-STRING.

| | PPI$_{>50}$ | | PPI$_{>100}$ | |
| | ROC-AUC | Accuracy | ROC-AUC | Accuracy |
|---|---|---|---|---|
| ERM | $74.11 \pm 0.35\%$ | $71.15 \pm 0.43\%$ | $71.91 \pm 0.24\%$ | $70.39 \pm 0.36\%$ |
| GroupDRO | $73.98 \pm 0.25\%$ | $69.59 \pm 0.56\%$ | $71.55 \pm 0.59\%$ | $67.00 \pm 0.85\%$ |
| IRM | $52.71 \pm 0.50\%$ | $64.30 \pm 0.02\%$ | $51.73 \pm 1.54\%$ | $63.16 \pm 1.82\%$ |
| IB-IRM | $52.12 \pm 0.91\%$ | $63.57 \pm 0.21\%$ | $52.33 \pm 1.06\%$ | $63.39 \pm 1.35\%$ |
| IB-ERM | $74.69 \pm 0.14\%$ | $71.47 \pm 0.43\%$ | $\underline{73.32 \pm 0.21\%}$ | $71.18 \pm 0.27\%$ |
| V-REx | $71.46 \pm 1.47\%$ | $71.33 \pm 0.90\%$ | $69.37 \pm 0.85\%$ | $70.93 \pm 1.06\%$ |
| DANN | $73.49 \pm 0.45\%$ | $70.74 \pm 0.38\%$ | $72.22 \pm 0.10\%$ | $70.41 \pm 0.25\%$ |
| CORAL | $\underline{75.42 \pm 0.15\%}$ | $71.71 \pm 0.34\%$ | $73.10 \pm 0.14\%$ | $70.88 \pm 0.10\%$ |
| MMD | $75.11 \pm 0.27\%$ | $71.57 \pm 0.40\%$ | $73.30 \pm 0.50\%$ | $71.15 \pm 0.63\%$ |
| RSC | $74.83 \pm 0.68\%$ | $71.20 \pm 0.34\%$ | $72.47 \pm 0.38\%$ | $70.77 \pm 0.50\%$ |
| CAD | $75.17 \pm 0.64\%$ | $71.97 \pm 0.83\%$ | $72.92 \pm 0.39\%$ | $\underline{71.34 \pm 0.46\%}$ |
| SelfReg | $\underline{75.42 \pm 0.42\%}$ | $70.34 \pm 0.57\%$ | $72.63 \pm 0.71\%$ | $69.14 \pm 0.90\%$ |
| Mixup | $74.40 \pm 0.54\%$ | $71.39 \pm 0.29\%$ | $71.31 \pm 1.06\%$ | $70.29 \pm 0.15\%$ |
| LISA | $74.30 \pm 0.59\%$ | $71.72 \pm 0.66\%$ | $71.45 \pm 0.44\%$ | $70.37 \pm 0.29\%$ |
| MAT | $74.73 \pm 0.30\%$ | $\underline{72.03 \pm 0.51\%}$ | $72.07 \pm 0.81\%$ | $71.09 \pm 0.44\%$ |
| AdaGraph | $74.02 \pm 0.42\%$ | $71.09 \pm 0.16\%$ | $72.10 \pm 0.06\%$ | $70.75 \pm 0.56\%$ |
| RaMoE | $74.99 \pm 0.22\%$ | $71.23 \pm 0.41\%$ | $71.48 \pm 0.49\%$ | $70.60 \pm 0.71\%$ |
| mDSDI | $75.09 \pm 0.47\%$ | $71.50 \pm 0.40\%$ | $71.23 \pm 0.69\%$ | $70.26 \pm 0.92\%$ |
| ADDAR | $74.55 \pm 0.54\%$ | $71.16 \pm 0.79\%$ | $71.93 \pm 0.33\%$ | $70.95 \pm 0.89\%$ |
| GRDA | $75.01 \pm 0.68\%$ | $71.47 \pm 0.91\%$ | $73.57 \pm 0.38\%$ | $70.44 \pm 0.29\%$ |
| DRM | $74.01 \pm 0.63\%$ | $71.68 \pm 0.29\%$ | $71.68 \pm 0.61\%$ | $70.55 \pm 0.82\%$ |
| LLE | $74.34 \pm 0.48\%$ | $71.26 \pm 0.17\%$ | $72.41 \pm 0.76\%$ | $70.61 \pm 0.59\%$ |
| DDN | $75.17 \pm 0.61\%$ | $71.75 \pm 0.52\%$ | $72.71 \pm 0.59\%$ | $71.02 \pm 0.27\%$ |
| TRO | $74.85 \pm 0.27\%$ | $71.53 \pm 0.16\%$ | $72.49 \pm 0.36\%$ | $70.88 \pm 0.51\%$ |
| **D$^3$G (ours)** | $\mathbf{78.67 \pm 0.16\%}$ | $\mathbf{74.25 \pm 0.34\%}$ | $\mathbf{77.24 \pm 0.30\%}$ | $\mathbf{73.50 \pm 0.24\%}$ |

Table 9: Full results of incorporating domain relations into the domain-specific learning.

| Model | | TPT-48 (MSE ↓) | | FMoW (Worst Acc. ↑) | | ChEMBL (ROC-AUC ↑) | |
| | | N (24) → S (24) | E (24) → W (24) | FMoW-Asia | FMoW-WILDS | PPI$_{>50}$ | PPI$_{>100}$ |
|---|---|---|---|---|---|---|---|
| DRM | | $0.603 \pm 0.041$ | $0.467 \pm 0.047$ | $26.37 \pm 1.19\%$ | $35.83 \pm 1.00\%$ | $74.01 \pm 0.63\%$ | $71.68 \pm 0.61\%$ |
| | + Fixed | $0.551 \pm 0.067$ | $0.371 \pm 0.033$ | $27.62 \pm 0.42\%$ | $37.58 \pm 0.67\%$ | $76.35 \pm 0.77\%$ | $74.32 \pm 0.47\%$ |
| LLE | | $0.571 \pm 0.038$ | $0.557 \pm 0.027$ | $25.22 \pm 2.33\%$ | $36.39 \pm 0.76\%$ | $74.34 \pm 0.48\%$ | $72.41 \pm 0.76\%$ |
| | + Fixed | $0.519 \pm 0.072$ | $0.417 \pm 0.049$ | $26.91 \pm 1.05\%$ | $37.11 \pm 0.21\%$ | $76.27 \pm 0.69\%$ | $74.81 \pm 0.36\%$ |
| DDN | | $0.537 \pm 0.024$ | $0.601 \pm 0.038$ | $27.89 \pm 0.95\%$ | $35.13 \pm 0.62\%$ | $75.17 \pm 0.61\%$ | $72.71 \pm 0.59\%$ |
| | + Fixed | $0.529 \pm 0.017$ | $0.433 \pm 0.052$ | $26.77 \pm 1.72\%$ | $37.25 \pm 0.44\%$ | $76.91 \pm 0.50\%$ | $75.57 \pm 0.68\%$ |
| **D$^3$G** | | $\mathbf{0.342 \pm 0.019}$ | $\mathbf{0.236 \pm 0.063}$ | $\mathbf{28.12 \pm 0.28\%}$ | $\mathbf{39.47 \pm 0.57\%}$ | $\mathbf{78.67 \pm 0.16\%}$ | $\mathbf{77.24 \pm 0.30\%}$ |

The results, as presented in Table 13, demonstrate that our method's variant outperforms ERM in 5 out of 7 settings, although it still lags behind D$^3$G. This suggests that the model architecture and inference algorithm can be adapted to learn domains and domain relations simultaneously in an end-to-end manner. Moreover, the appropriate utilization of domain meta-data and domain relations significantly improves performance.

**D$^3$G w/o domain meta-data**: In situations where domain meta-data is unavailable, we learn domain relations directly from the data. To achieve this, we compute the fixed relation using optimal transport Alvarez-Melis and Fusi (2020). For the learnable part, we utilize the average of features from each domain as input to the two-layer neural network $g$.

According to Table 13, the findings indicate that this variant exhibits inferior performance compared to the one with domain meta-data. However, it still outperforms the best baselines on most datasets. These outcomes suggest that the model design of D$^3$G inherently enhances generalization capabilities, which are further augmented by the use of domain meta-data.

Table 10: Comparison of using different relations. The results on FMoW and ChEMBL-STRING are reported. In this case, when no relations are used, we take the average of predictions across all domains.

| Fixed relations | Learned relations | FMoW (Worst Acc. ↑) | | ChEMBL-STRING (ROC-AUC ↑) | |
|---|---|---|---|---|---|
| | | FMoW-Asia | FMoW-WILDS | $PPI_{>50}$ | $PPI_{>100}$ |
| | | $26.93 \pm 0.47\%$ | $35.32 \pm 0.66\%$ | $76.17 \pm 0.21\%$ | $73.38 \pm 0.13\%$ |
| ✓ | | $27.43 \pm 0.41\%$ | $39.37 \pm 0.34\%$ | $77.66 \pm 0.32\%$ | $76.59 \pm 0.40\%$ |
| | ✓ | $21.18 \pm 2.30\%$ | $36.41 \pm 1.09\%$ | $77.09 \pm 0.94\%$ | $75.57 \pm 1.20\%$ |
| ✓ | ✓ | $\mathbf{28.12 \pm 0.28\%}$ | $\mathbf{39.47 \pm 0.57\%}$ | $\mathbf{78.67 \pm 0.16\%}$ | $\mathbf{77.24 \pm 0.30\%}$ |

Table 11: Full results of comparison between $D^3G$ with domain-specific fine-tuning.

| Model | FMoW (Worst Acc. ↑) | | ChEMBL (ROC-AUC ↑) | |
|---|---|---|---|---|
| | FMoW-Asia | FMoW-WILDS | $PPI_{>50}$ | $PPI_{>100}$ |
| ERM | $26.05 \pm 3.84\%$ | $34.87 \pm 0.41\%$ | $74.11 \pm 0.35\%$ | $71.91 \pm 0.24\%$ |
| CORAL | $25.87 \pm 1.97\%$ | $36.53 \pm 0.15\%$ | $75.42 \pm 0.15\%$ | $73.10 \pm 0.14\%$ |
| RW-FT | $27.03 \pm 1.03\%$ | $36.39 \pm 1.28\%$ | $76.31 \pm 0.35\%$ | $74.30 \pm 0.40\%$ |
| $\mathbf{D^3G}$ | $\mathbf{28.12 \pm 0.28\%}$ | $\mathbf{39.47 \pm 0.57\%}$ | $\mathbf{78.67 \pm 0.16\%}$ | $\mathbf{77.24 \pm 0.30\%}$ |

# I LIMITATIONS

While our work provides theoretical results, it is important to acknowledge the limitations of our approach. We rely on certain assumptions, and therefore, $D^3G$ does not guarantee a proven enhancement in out-of-domain generalization in all scenarios. However, we are committed to expanding our theoretical results in future research. Furthermore, it is worth noting that $D^3G$ extracts domain relations from domain meta-data. However, the performance can be affected in cases where the domain meta-data is of inferior quality. In our future work, we will address this issue by providing detailed insights on obtaining high-quality domain meta-data.

Table 12: Full results of comparison between $D^3G$ with using separate feature extractors for each training domain.

| Model | FMoW (Worst Acc. ↑) | | ChEMBL (ROC-AUC ↑) | |
|---|---|---|---|---|
| | FMoW-Asia | FMoW-WILDS | $PPI_{>50}$ | $PPI_{>100}$ |
| Using separate feature extractors | $23.62 \pm 0.47\%$ | $35.44 \pm 0.58\%$ | $76.11 \pm 0.31\%$ | $74.19 \pm 0.74\%$ |
| **$D^3G$** | **$28.12 \pm 0.28\%$** | **$39.47 \pm 0.57\%$** | **$78.67 \pm 0.16\%$** | **$77.24 \pm 0.30\%$** |

Table 13: Full results of comparison between $D^3G$ with its variant that use limited domain information.

| Model | FMoW (Worst Acc. ↑) | | ChEMBL (ROC-AUC ↑) | |
|---|---|---|---|---|
| | FMoW-Asia | FMoW-WILDS | $PPI_{>50}$ | $PPI_{>100}$ |
| $D^3G$ w/o domain information | $25.46 \pm 1.35\%$ | $36.06 \pm 0.39\%$ | $74.94 \pm 0.79\%$ | $73.38 \pm 0.91\%$ |
| $D^3G$ w/o domain meta-data | $27.17 \pm 1.09\%$ | $37.82 \pm 0.31\%$ | $76.65 \pm 0.22\%$ | $75.01 \pm 0.36\%$ |
| **$D^3G$** | **$28.12 \pm 0.28\%$** | **$39.47 \pm 0.57\%$** | **$78.67 \pm 0.16\%$** | **$77.24 \pm 0.30\%$** |

