# OpenReview forum: "Improving Domain Generalization with Domain Relations"
_ICLR.cc/2024/Conference — ICLR 2024 spotlight_

### Official Review · Reviewer_u1oN · 2023-10-30

**Soundness:** 2 fair
**Presentation:** 3 good
**Contribution:** 2 fair
**Rating:** 6
**Confidence:** 4

**Summary:**

This paper proposed D3G, which in contrast to previous approaches, utilizes domain metadata to create domain-specific models. During training, it learns domain-specific functions, and during testing, it reweights them based on domain relations, enhancing model adaptability using directly acquired domain metadata.

**Strengths:**

- The paper is well-written and easy to follow. The authors use appropriate notation and equations where necessary.
- This paper utilizes domain meta-data to extract relations between domains.
- The domain relations are learned rather than fixed.

**Weaknesses:**

(1) In section 3.2, it is mentioned that both $m_i$ and $a_{ij}^{g}$ are derived from domain metadata, but the specific relationship or distinction between them is not clearly explained. The experimental results suggest that $a_{ij}^{g}$ is extracted from $m_i$, yet this section lacks explicit clarification.

(2) It is unclear whether the two layer neural network $g$ should be trained or directly be used from other previous works. It is mentioned that weight vectors $w_r$ are learnable vectors. Are these weights and the neural network trained by the consistency loss?

(3) The consistency loss implies that each head is trained on all the domains with different weights. This is similar to the case where each head is trained on all the domains with a weighted combination of the losses of all the domains. For the predictor $f^{(d)}$, the loss is in this form: $l(\frac{\sum_j^{N^{tr}} a_{dj} f^{(d)}(x^j)}{\sum_j^{N_{tr}} a_{dj}})$. Would this loss also help the learning of domain-specific models?

(4) Suppose that each head is trained on all the domains by the loss function (2). In this case, regardless of the values of $a_{ij}$, if both the supervised loss and consistency loss reach their minimum, then the learning of $a_{ij}$ would be useless. Does the proposed method face this issue?

(5) In the experiment section, one baseline of the proposed method would be training each head on all the training domains by supervised loss and averaging the predictions of these heads as the test prediction.

**Questions:**

Please refer to the Weakness.

---

> ### Author Response · Authors · 2023-11-21
> **Response to Reviewer u1oN (1/2)**
>
> Thank you so much for the insightful and valuable comments! They are very helpful for further improving the clarity and quality of our paper. We'll revise our manuscript in the next version to address all of your concerns.
>
> **Q1**: Section 3.2 lacks explicit clarification of the specific relationship or distinction between domain meta-data and domain relations.
>
> **A1**: Thank you for pointing that out. The fixed domain relation $a_{ij}^g$ can be derived either from domain meta-data $m_i$ or directly captured by domain experts. In the former case, for instance, in spatial distribution shift scenarios like FMoW, we can use geographical information (e.g., longitude and latitude), which may be included in the meta-data $m_i$ or $m_j$, to establish geographical proximity relations. In the latter case, as an example, in drug-target interaction prediction, we employ a fixed relation represented by the protein-protein graph, while $m_i$ represents the protein structure. We have clarified these points in the updated paper in Section 3.2.
>
> ---
>
> **Q2**: It is unclear whether the two layer neural network $g$ should be trained or directly be used from other previous works.
>
> **A2**: In this paper, the final domain relation $a_{ij}$ contains the learned relation $a_{ij}^l$, which is learned from the domain meta-data (Eqn. (5)) with the two layer neural network. The final domain relation $a_{ij}$ is reflected in the consistency training loss. Thus, the two-layer neural network can only be optimized during the training process only.
>
> ---
>
> **Q3**: The similarity between the consistency loss and the case where each head is trained on all the domains with a weighted combination of the losses of all the domains. Would this loss also help the learning of domain-specific models?
>
> **A3**: Yes, this loss is similar to our consistency loss and would help the learning of domain-specific models. We evaluate the performance of D$^3$G with this loss, and present the results in Table R1.
>
> **Table R1**: Performance comparison between D$^3$G and its variant with losses from all the domains.
>
> |      |   DG-15 (Avg. Acc. $\uparrow$)    |        TPT-48 (MSE $\downarrow$)        |                           | FMoW (Worst Acc. $\uparrow$) |              | ChEMBL-STRING (ROC-AUC $\uparrow$) |              |
> | :--------------- | :-----------------: | :-----------------------: | :-----------------------: | :-------------------: | :-------------------: | :--------------------: | :-------------------: |
> |                  |                     | N(24) $\rightarrow$ S(24) | E(24) $\rightarrow$ W(24) |       FMoW-Asia       |      FMoW-WILDS       |         PPI>50         |        PPI>100        |
> |         D$^3$G with losses from all the domains        |   71.8 $\pm$ 1.8%   |     0.367 $\pm$ 0.041     |     0.249 $\pm$ 0.037     |   27.99 $\pm$ 0.67%   |   38.75 $\pm$ 0.51%   |   78.40 $\pm$ 0.11%   |   76.52 $\pm$ 0.49%   |
> | **D$^3$G(Ours)** | **77.5 $\pm$ 2.5%** |   **0.342 $\pm$ 0.019**   |   **0.236 $\pm$ 0.063**   | **28.12 $\pm$ 0.28%** | **39.47 $\pm$ 0.57%** | **78.67 $\pm$ 0.16%**  | **77.24 $\pm$ 0.30%** |
>
> The above results show that D$^3$G, utilizing weighted losses mentioned by Reviewer u1oN, performs well across all datasets since our original two-term loss (predictive loss + consistency loss) approximates a weighted loss form. Our original loss essentially upweights the corresponding domain of each head more significantly using a separate predictive loss term, leading to more improvement over the new variant.

---

> ### Author Response · Authors · 2023-11-21
> **Response to Reviewer u1oN (2/2)**
>
> **Q4**: Subquestion 1: Suppose that each head is trained on all the domains by the loss function (2). In this case, regardless of the values of $a_{ij}$ if both the supervised loss and consistency loss reach their minimum, then the learning of $a_{ij}$would be useless. Does the proposed method face this issue?
>
> Subquestion 2: Results of a new baseline: training each head on all the training domains by supervised loss and averaging the predictions of these heads as the test prediction.
>
> **A4**: If each head is trained on all domains and results in minimal predictive loss and consistency loss, it implies that all domains are equally important in this problem, and it suggests that all $a_{ij}$ are identical. This can be considered a special case of our developed method. However, in real-world scenarios, encountering this situation is rare. This issue can be verified through the following experiment:
>
> As suggested by the reviewer u1oN, we train each head on all the training domains using supervised loss and average the predictions of these heads as the test prediction. This experiment has been conducted in our original submission (see the first row in Table 3 in our paper, where no relations are used). We also present the corresponding results in Table R2.
>
> **Table R2**: Performance comparison between D$^3$G and its variant with no related domains.
>
>
> |      |   FMoW (Worst Acc. $\uparrow$)    |                                     | ChEMBL-STRING (ROC-AUC $\uparrow$) |                       |
> |:-----------------|:----------:|:----------:|:---------:|:---------:|
> |                  | FMoW-Asia | FMoW-WILDS | PPI>50 |        PPI>100        |
> | D3G with no related domains |   26.93 $\pm$ 0.47%   |          35.32 $\pm$ 0.66%          |        76.17 $\pm$ 0.21%        |   73.38 $\pm$ 0.13%   |
> | **D3G (Ours)** | **28.12 $\pm$ 0.28%** |        **39.47 $\pm$ 0.57%**        |      **78.67 $\pm$ 0.16%**      | **77.24 $\pm$ 0.30%** |
>
> The results reveal that this baseline slightly outperforms ERM, yet significantly underperforms compared to our D$^3$G. This indicates the importance of using uneven domain relation to weight losses and predictions from different domains/heads.

---

> > ### Comment · Reviewer_u1oN · 2023-11-23
> >
> > Thank you for your response. I keep the current score.

---

### Official Review · Reviewer_Htjr · 2023-11-01

**Soundness:** 3 good
**Presentation:** 3 good
**Contribution:** 3 good
**Rating:** 6
**Confidence:** 3

**Summary:**

This paper proposed a new approach for out-of-distribution generalization named as D$^3$G. It leverages domain relations estimated from domain metadata to learn training domain specific models and the ensemble of training-domain-specific functions. Under some assumptions, for example the domain relations can accurately capture the model similarity between domains, the authors proved that the proposed method can generalize better to out-of-domain samples compared to the traditional averaging approach. Empirical results on both synthetic and real-world datasets show that D$^3$G surpassing the performance of traditional averaging methods and some other baseline methods.

**Strengths:**

- Originality: good. Though learning domain specific classification head has been proposed in exiting works, it is novel to ensemble domain specific classifiers by weights learned from domain meta-data.
- Significance: the proposed method is simple but shown improved performance on several tasks.
- The paper is overall well written and easy to follow.
- Experiment setups are introduced in detail.

**Weaknesses:**

- The robustness and generality of the proposed method is unclear. Practically, it is unclear how to design a good similarity definition for metadata of different tasks. The current way of constructing the fixed relations requires specific and expertise knowledge on each task. Nevertheless a learning approach is proposed, the ablation study results in Table 10 show that the learned relations can be less helpful on some tasks.
- The limitations of this work are not fully discussed. For example, the generality of the assumptions in the theoretical analysis part.

**Questions:**

Please refer to the concerns in the "weaknesses" part.

---

> ### Author Response · Authors · 2023-11-21
> **Response to Reviewer Htjr**
>
> Thank you so much for the insightful and valuable comments! They are very helpful for further improving the clarity and quality of our paper. We'll revise our manuscript in the next version to address all of your concerns.
>
> **Q1**: The robustness and generality of the proposed method is unclear. Practically, it is unclear how to design a good similarity definition for metadata of different tasks.
>
> **A1**: In our experiments, we use cosine similarity for DG-15, TPT-48, and FMoW, and apply PPI scores for biomedical data. Consequently, cosine similarity serves as a reliable backup for defining similarity when domain meta-data is featured.
>
> In situations where domain meta-data is discrete (e.g., label or index) and cosine similarity cannot be applied, we can directly learn domain relations directly from the data. To do this, we compute the fixed relation across different domains by applying optimal transport on the corresponding data [1]. For the learnable part, we utilize the average of features from each domain as input to the two-layer neural network $g$. We report the results in Table R1. We’ve added these results in Appendix H.3 in the updated paper.
>
> **Table R1**: Performance comparison between D$^3$G and its variant that inly utilize domain ids, along with reporting the best baseline performance for each dataset.
>
> |      |   DG-15 (Avg. Acc. $\uparrow$)    |        TPT-48 (MSE $\downarrow$)        |                           | FMoW (Worst Acc. $\uparrow$) |              | ChEMBL-STRING (ROC-AUC $\uparrow$) |              |
> | :--------------- | :-----------------: | :-----------------------: | :-----------------------: | :-------------------: | :-------------------: | :--------------------: | :-------------------: |
> |                  |                     | N(24) $\rightarrow$ S(24) | E(24) $\rightarrow$ W(24) |       FMoW-Asia       |      FMoW-WILDS       |         PPI>50         |        PPI>100        |
> |         Best baseline        |   66.2 $\pm$ 2.0%   |     0.371 $\pm$ 0.054     |     0.262 $\pm$ 0.034     |   26.99 $\pm$ 1.27%   |   37.64 $\pm$ 0.92%   | 75.42 $\pm$ 0.42% | 73.57 $\pm$ 0.21% |
> |         D$^3$G w/o domain meta-data        |   70.2 $\pm$ 3.7%   |     0.381 $\pm$ 0.071     |     0.325 $\pm$ 0.029     |   27.17 $\pm$ 1.09%   |   37.82 $\pm$ 0.31%   |   76.65 $\pm$ 0.22%   |   75.01 $\pm$ 0.36%   |
> | **D$^3$G (Ours)** | **77.5 $\pm$ 2.5%** |   **0.342 $\pm$ 0.019**   |   **0.236 $\pm$ 0.063**   | **28.12 $\pm$ 0.28%** | **39.47 $\pm$ 0.57%** | **78.67 $\pm$ 0.16%**  | **77.24 $\pm$ 0.30%** |
>
> The findings indicate that this variant exhibits inferior performance compared to the one with domain meta-data, however, it still outperforms the best baselines on most datasets. These outcomes suggest that we can also design a relatively good similarity definition for meta-data of different tasks, even with only labels or indices to distinguish different domains. This further confirms the generality of D$^3$G.
>
> ---
>
> **Q2**: The limitations of this work are not fully discussed. For example, the generality of the assumptions in the theoretical analysis part.
>
> **A2**: We’ve discussed the limitations of our work in our original submission (see Appendix I). We’ve added the reference in the conclusion (see “We discussed our limitations in Appendix I”).
>
> ---
>
> **References**:
>
> [1] David Alvarez-Melis and Nicolò Fusi. Geometric dataset distances via optimal transport. ArXiv, abs/2002.02923, 2020.

---

### Official Review · Reviewer_79WW · 2023-11-01

**Soundness:** 3 good
**Presentation:** 2 fair
**Contribution:** 2 fair
**Rating:** 6
**Confidence:** 3

**Summary:**

This paper presents a novel method called D3G for tackling the issue of domain shifts in real-world machine learning scenarios. The approach leverages the connections between different domains to enhance the model’s robustness and employs a domain-relationship aware weighting system for each test domain.

**Strengths:**

The authors provide a theoretical proof that using domain relations to reweight a specific function of the training domain can obtain stronger extraterritorial generalization.

**Weaknesses:**

The novelty is limited, perhaps the authors have not sufficiently explained the differences between their work and existing methods.

The authors state that " Unlike prior works that rely on ensemble models to address the underspecification problem and improve out-of-distribution robustness, our proposed D3G takes a conceptually different approach by constructing domain-specific models." However, the  MoE[1] constructed domain-specific models and they adopted ensemble models to improve out-of-distribution robustness.

The meta-data is key of domain-relation in this work. However, there is no clear definition of meta-data, how to obtain meta-data, and why the meta-data work is lack of detailed elaboration and analysis.

How the consistency loss address the challenge of limited training data in certain domains?

The organizational of the method section needs to be adjusted. The description and calculation of the domain relationship "a" in advance make the method part more clear.

It is recommended to add the venue and year of the comparison method to the table in the experiment section.


[1]Dai, Yongxing, et al. "Generalizable person re-identification with relevance-aware mixture of experts." Proceedings of the IEEE/CVF Conference on Computer Vision and Pattern Recognition. 2021.

**Questions:**

See the weakness.

---

> ### Author Response · Authors · 2023-11-21
> **Response to Reviewer 79WW (1/2)**
>
> Thank you for your valuable comments and suggestions. We have carefully revised our paper based on your comments. Our responses to your questions are detailed below. We would greatly appreciate your input on whether our revisions address your concerns.
>
> **Q1**: The novelty is limited, perhaps the authors have not sufficiently explained the differences between their work and existing methods.
>
> **A1**: We appreciate the reviewer's concern regarding the novelty of our method in comparison with existing methods. While it is true that training specific models for each domain and combining them is similar to the mixture-of-experts approach, the major contribution of our work is the use of domain meta-data to generate more accurate domain relations. This is crucial since it is challenging to directly extract domain specific information and learn domain relational information from training samples in real-world applications. We also introduce a consistency loss to mitigate the impact of data size imbalance across domains. Both theoretical and empirical results demonstrate the effectiveness of our approach. A detailed comparison between D$^3$G and MoE is presented in the response (A2) of your question 2 (Q2).
>
> In general, our D$^3$G pioneers the importance of accurate domain relations for improved generalization performance. Instead, though mentioning domain relation/distance/similarity, these previous methods focus on learning domain-specific functions and devote less effort to acquiring accurate relations. As a result, their learned weight does not reflect accurate domain relations, leading to a worse performance. Unlike these, we tackle this challenge by incorporating domain meta-data, which is a fresh perspective that sets our work apart from existing literature.
>
> ---
>
> **Q2**: Comparison with MoE[1]
>
> **A2**: Our proposed method shares some similarities with multi-head network structures in MoE [1], but our D$^3$G can generate more accurate domain relations by using domain meta-data. To evaluate the performance of our method, we compared it with MoE on all datasets. The result is presented in Table R1, and we’ve added these results in Figure 2 and Table 1 in the updated paper.. According to the results, we observe that D$^3$G still outperforms MoE and shows its effectiveness. We've included these results in our main paper in the updated version. Besides MoE, in our paper, we also compare several multi-head models without using domain meta-data such as DRM [2], LLE [3], and DDN [4]. Please refer to the paper for these results.
>
> **Table R1**: Performance comparison between D$^3$G and MoE on all datasets.
>
> |      |   DG-15 (Avg. Acc. $\uparrow$)    |        TPT-48 (MSE $\downarrow$)        |                           | FMoW (Worst Acc. $\uparrow$) |              | ChEMBL-STRING (ROC-AUC $\uparrow$) |              |
> |:-----------------|:----------:|:----------:|:---------:|:---------:|:----------------:|:----------------:|:----------------:|
> |                  |                       | N(24) $\rightarrow$ S(24) | E(24) $\rightarrow$ W(24) |    FMoW-Asia     |  FMoW-WILDS  |         PPI>50         |   PPI>100    |
> | MoE              |   53.7 $\pm$ 1.9%   |     0.372 $\pm$ 0.035     |     0.311 $\pm$ 0.060     |   26.65 $\pm$ 0.46%   |   36.51 $\pm$ 0.71%   |   74.99 $\pm$ 0.22%   |   71.48 $\pm$ 0.49%   |
> | **D$^3$G (Ours)** | **77.5 $\pm$ 2.5%** |   **0.342 $\pm$ 0.019**   |   **0.236 $\pm$ 0.063**   | **28.12 $\pm$ 0.28%** | **39.47 $\pm$ 0.57%** | **78.67 $\pm$ 0.16%**  | **77.24 $\pm$ 0.30%** |
>
> ---
>
> **Q3**: The meta-data is the key of domain-relation in this work. However, there is no clear definition of meta-data, how to obtain meta-data, and why the meta-data work is lacking detailed elaboration and analysis.
>
> **A3**: The domain meta-data can be collected from various sources, depending on the application. For example, in applications related to spatial distribution shift (e.g., land usage prediction in FMoW), geographical information such as longitude and latitude can be used as domain relations. In the context of drug discovery, where different proteins are treated as different domains, we can use protein structures as domain meta-data. Since the domain meta-data includes abundant domain information, it can be used to accurately measure domain relations, thereby enabling more precise domain generalization and knowledge transfer from training domains to test domains.

---

> > ### Comment · Reviewer_79WW · 2023-11-21
> >
> > The authors have addressed some of my concerns in their responses. Additionally, I would like to inquire whether this metadata is akin to the introduction of prior knowledge. In the context of various tasks, is the metadata extracted automatically by the algorithm, or does it require manual intervention?

---

> ### Author Response · Authors · 2023-11-21
> **Response to Reviewer 79WW (2/2)**
>
> **Q4**: How does consistency loss address the challenge of limited training data in certain domains?
>
>
> **A4**: The consistency loss is utilized to establish connections between domains and facilitate training on data insufficient domains. Throughout the training process, for all examples from domain $d$, we use these examples and employ the overall loss not only to update its domain-specific function $f^{(d)}$ but also to influence other domain-specific functions with the consistency loss. Consequently, the utilization of data from one domain effectively updates all parameters, implying that domain-specific models in data-scarce domains are optimized not only by their respective domain data but also by other domains, particularly those that are similar. For example, if training domain $d$ is more similar to training domain $d-1$ than to domain $d+1$, it will have a stronger influence on the optimization of the function $f^{(d-1)}$.
>
> ---
>
> **References**:
>
> [1] Dai, Yongxing, et al. "Generalizable person re-identification with relevance-aware mixture of experts." Proceedings of the IEEE/CVF Conference on Computer Vision and Pattern Recognition. 2021.
>
> [2] Yi-Fan Zhang, Han Zhang, Jindong Wang, Zhang Zhang, B. Yu, Liangdao Wang, Dacheng Tao, and Xingxu Xie. Domain-specific risk minimization. ArXiv, abs/2208.08661, 2022b.
>
> [3] Zhiheng Li, Ivan Evtimov, Albert Gordo, Caner Hazirbas, Tal Hassner, Cristian Canton Ferrer, Chenliang Xu, and Mark Ibrahim. A whac-a-mole dilemma: Shortcuts come in multiples where mitigating one amplifies others. June 2023. URL https://arxiv.org/abs/2212.04825.
>
> [4] Daoan Zhang, Mingkai Chen, Chenming Li, Lingyun Huang, and Jianguo Zhang. Aggregation of disentanglement: Reconsidering domain variations in domain generalization. ArXiv, abs/2302.02350, 2023.

---

> ### Author Response · Authors · 2023-11-21
> **Additional Response to Reviewer 79WW**
>
> Dear Reviewer 79WW,
>
> Thank you very much for your prompt response.
>
> Yes, the metadata is akin to the prior knowledge, which is not extracted by the algorithm. Most often, domain meta-data is obtained as a by-product during the collection of test data, which then serves as the primary criteria for classifying domains. This meta-data typically manifests as an inherent property of the testing domain, such as latitude and longitude in geographic data or PPI scores in biomedical data. In most scenarios, obtaining meta-data for each testing domain is not overly expensive since the number of testing domains is relatively small.
>
> Nevertheless, In situations where domain metadata and the domain similarity function are unavailable, we can derive domain distances directly from the data. To do this, we compute the fixed relation using optimal transport [1]. For the learnable part, we utilize the average of features from each domain as input to the two-layer neural network $g$. We report the results in Table R2.
>
> **Table R2**: Performance comparison between D$^3$G and its variant that does not utilize domain meta-data, along with reporting the best baseline performance for each dataset.
>
> |      |   DG-15 (Avg. Acc. $\uparrow$)    |        TPT-48 (MSE $\downarrow$)        |                           | FMoW (Worst Acc. $\uparrow$) |              | ChEMBL-STRING (ROC-AUC $\uparrow$) |              |
> | :--------------- | :-----------------: | :-----------------------: | :-----------------------: | :-------------------: | :-------------------: | :--------------------: | :-------------------: |
> |                  |                     | N(24) $\rightarrow$ S(24) | E(24) $\rightarrow$ W(24) |       FMoW-Asia       |      FMoW-WILDS       |         PPI>50         |        PPI>100        |
> |         Best baseline        |   66.2 $\pm$ 2.0%   |     0.371 $\pm$ 0.054     |     0.262 $\pm$ 0.034     |   26.99 $\pm$ 1.27%   |   37.64 $\pm$ 0.92%   | 75.42 $\pm$ 0.42% | 73.57 $\pm$ 0.21% |
> |         D$^3$G w/o domain meta-data        |   70.2 $\pm$ 3.7%   |     0.381 $\pm$ 0.071     |     0.325 $\pm$ 0.029     |   27.17 $\pm$ 1.09%   |   37.82 $\pm$ 0.31%   |   76.65 $\pm$ 0.22%   |   75.01 $\pm$ 0.36%   |
> | **D$^3$G (Ours)** | **77.5 $\pm$ 2.5%** |   **0.342 $\pm$ 0.019**   |   **0.236 $\pm$ 0.063**   | **28.12 $\pm$ 0.28%** | **39.47 $\pm$ 0.57%** | **78.67 $\pm$ 0.16%**  | **77.24 $\pm$ 0.30%** |
>
> The findings indicate that this variant exhibits inferior performance compared to the one with domain meta-data, however, it still outperforms the best baselines on most datasets. These outcomes suggest that D$^3$G's model design inherently enhances generalization capabilities, which are further augmented by the use of domain meta-data. We’ve added these results in Appendix H.4 in the updated paper.

---

> > ### Comment · Reviewer_79WW · 2023-11-21
> >
> > Thanks for the reply, I will raise my score.

---

> > > ### Author Response · Authors · 2023-11-21
> > > **Thank you for raising your score**
> > >
> > > Dear Reviewer 79WW,
> > >
> > > We are happy to see our response addresses your questions. Thank you for raising your rating.

---

### Official Review · Reviewer_bK1N · 2023-11-02

**Soundness:** 4 excellent
**Presentation:** 4 excellent
**Contribution:** 4 excellent
**Rating:** 8
**Confidence:** 4

**Summary:**

The paper addresses the domain shift problem and proposes a new approach called D3G to this problem. Instead of learning a single model from multiple source domains, the proposed method learns domain-specific models and infer a test domain specific model by exploiting the domain relations. The test domain model is a weighted combination of multiple source domain models whilst the weights are learned from the domain metadata. Theoretic and empirical analyses have been made and experimental results demonstrate the superiority of the proposed approach to many state-of-the-art counterparts.

**Strengths:**

-- The approach distincts from most existing ones in that it learns domain-specific models instead of a unified model.

-- The method is clearly presented and theoretic analysis has been made to clarify why it should work for domain generalization.

-- The proposed approach is applicable to many real-world applications as demonstrated in the experiments. The superior performance makes a significant difference to practical problems of this type.

-- Ablation studies have been conducted to demonstrate the effectiveness of each component of the proposed approach.

**Weaknesses:**

-- The comparison of the proposed method with ensemble models of existing approaches should have been given.

-- There exist some typos/language issues, e.g., "Eqn. equation x..."; "We using weighted...";

**Questions:**

1. Does it work when there is no domain metadata available? Can such domain relations be learned from the data themselves?
2. How the domain metadata are used in the comparative methods?

---

> ### Author Response · Authors · 2023-11-21
> **Response to Reviewer bK1N (1/2)**
>
> Thank you for your constructive comments and suggestions. We have revised our paper according to your comments. We respond to your questions below and would appreciate it if you could let us know if our response addresses your concerns.
>
> **Q1**: The comparison of the proposed method with ensemble models of existing approaches should have been given.
>
> **A1**: Our proposed method shares some similarities with ensemble methods, yet it is more closely aligned with multi-head network structures. As illustrated in Figure 1 in the paper, our approach employs a shared feature extractor and learns a collection of domain-specific heads. Consequently, in our paper, we compare several multi-head models without using domain meta-data, such as DRM [1], LLE [2], and DDN [3]. Our method consistently outperforms these baselines, highlighting the significant role of domain meta-data in generating accurate domain relations and enhancing generalization performance.
>
> Furthermore, we have also conducted an additional comparison between D$^3$G and one variant, which employs separate feature extractors for each training domain. The results are reported in Table R1 here.
>
> **Table R1**: Performance comparison between D$^3$G and using separate feature extractors for each training domain.
>
> |                  |  DG-15 (Avg. Acc. $\uparrow$)   |        TPT-48 (MSE $\downarrow$)        |                           |   FMoW (Worst Acc. $\uparrow$)    |                       | ChEMBL-STRING (ROC-AUC $\uparrow$) |                       |
> | :--------------- | :-----------------: | :-----------------------: | :-----------------------: | :-------------------: | :-------------------: | :--------------------: | :-------------------: |
> |                  |                     | N(24) $\rightarrow$ S(24) | E(24) $\rightarrow$ W(24) |       FMoW-Asia       |      FMoW-WILDS       |         PPI>50         |        PPI>100        |
> |         Using separate feature extractors        |   47.3 $\pm$ 3.7%   |     0.563 $\pm$ 0.071     |     0.519 $\pm$ 0.076     |   23.62 $\pm$ 0.47%   |   35.44 $\pm$ 0.58%   |   76.11 $\pm$ 0.31%    |   74.19 $\pm$ 0.74%   |
> | **D$^3$G(Ours)** | **77.5 $\pm$ 2.5%** |   **0.342 $\pm$ 0.019**   |   **0.236 $\pm$ 0.063**   | **28.12 $\pm$ 0.28%** | **39.47 $\pm$ 0.57%** | **78.67 $\pm$ 0.16%**  | **77.24 $\pm$ 0.30%** |
>
> According to the results, D$^3$G shows superior performance compared to using separate feature extractors for each training domain. This underscores the importance of employing a shared feature extractor to learn a universal representation, while domain-specific heads can further identify domain-specific features. Additionally, our D$^3$G model demonstrates superior efficiency compared to this variant since it utilizes a shared feature extractor while the variant needs $N^{tr}$ feature extractors. We’ve added these results in Appendix H.3 in the updated paper.
>
>
> ---
>
> **Q2**: There exist some typos/language issues, e.g., "Eqn. equation x..."; "We using weighted..."
>
> **A2**: Thanks for pointing that out. We’ve fixed these issues in the updated paper.

---

> ### Author Response · Authors · 2023-11-21
> **Response to Reviewer bK1N (2/2)**
>
> **Q3**: Does it work when there is no domain metadata available? Can such domain relations be learned from the data themselves?
>
> **A3**: Yes, we do an ablation study for this question by learning domain relations directly from the data. To do this, we compute the fixed relation using optimal transport [4]. For the learnable part, we utilize the average of features from each domain as input to the two-layer neural network $g$. We report the results in Table R2.
>
> **Table R2**: Performance comparison between D$^3$G and its variant that does not utilize domain meta-data, along with reporting the best baseline performance for each dataset.
>
> |                  |  DG-15 (Avg. Acc. $\uparrow$)   |        TPT-48 (MSE $\downarrow$)        |                           |   FMoW (Worst Acc. $\uparrow$)    |                       | ChEMBL-STRING (ROC-AUC $\uparrow$) |                       |
> | :--------------- | :-----------------: | :-----------------------: | :-----------------------: | :-------------------: | :-------------------: | :--------------------: | :-------------------: |
> |                  |                     | N(24) $\rightarrow$ S(24) | E(24) $\rightarrow$ W(24) |       FMoW-Asia       |      FMoW-WILDS       |         PPI>50         |        PPI>100        |
> |         Best baseline        |   66.2 $\pm$ 2.0%   |     0.371 $\pm$ 0.054     |     0.262 $\pm$ 0.034     |   26.99 $\pm$ 1.27%   |   37.64 $\pm$ 0.92%   | 75.42 $\pm$ 0.42% | 73.57 $\pm$ 0.21% |
> |         D$^3$G w/o domain meta-data        |   70.2 $\pm$ 3.7%   |     0.381 $\pm$ 0.071     |     0.325 $\pm$ 0.029     |   27.17 $\pm$ 1.09%   |   37.82 $\pm$ 0.31%   |   76.65 $\pm$ 0.22%   |   75.01 $\pm$ 0.36%   |
> | **D$^3$G(Ours)** | **77.5 $\pm$ 2.5%** |   **0.342 $\pm$ 0.019**   |   **0.236 $\pm$ 0.063**   | **28.12 $\pm$ 0.28%** | **39.47 $\pm$ 0.57%** | **78.67 $\pm$ 0.16%**  | **77.24 $\pm$ 0.30%** |
>
> The findings indicate that this variant exhibits inferior performance compared to the one with domain meta-data, however, it still outperforms the best baselines on most datasets. These outcomes suggest that D$^3$G's model design inherently enhances generalization capabilities, which are further augmented by the use of domain meta-data. We’ve added these results in Appendix H.4 in the updated paper.
>
> ---
>
> **Q4**: How are the domain metadata used in the comparative methods?
>
> **A4**: As we discussed in the caption of Table 1 (page 8) and in the first paragraph of experiment section, for all baselines, we incorporate domain meta-data as
> features during the training and test stages for fair comparison. In detail, we begin by converting domain meta-data into input embeddings, which are then passed through the two-layer neural network. Subsequently, we concatenate this intermediate domain feature with other features and pass this concat feature through another feature extractor. This procedure is consistently applied to all baselines.
>
> ---
>
> **References**:
>
> [1] Yi-Fan Zhang, Han Zhang, Jindong Wang, Zhang Zhang, B. Yu, Liangdao Wang, Dacheng Tao, and Xingxu Xie. Domain-specific risk minimization. ArXiv, abs/2208.08661, 2022b.
>
> [2] Zhiheng Li, Ivan Evtimov, Albert Gordo, Caner Hazirbas, Tal Hassner, Cristian Canton Ferrer, Chenliang Xu, and Mark Ibrahim. A whac-a-mole dilemma: Shortcuts come in multiples where mitigating one amplifies others. June 2023. URL https://arxiv.org/abs/2212.04825.
>
> [3] Daoan Zhang, Mingkai Chen, Chenming Li, Lingyun Huang, and Jianguo Zhang. Aggregation of disentanglement: Reconsidering domain variations in domain generalization. ArXiv, abs/2302.02350, 2023.
>
> [4] David Alvarez-Melis and Nicolò Fusi. Geometric dataset distances via optimal transport. ArXiv, abs/2002.02923, 2020.

---

### Official Review · Reviewer_2jGa · 2023-11-04

**Soundness:** 3 good
**Presentation:** 3 good
**Contribution:** 3 good
**Rating:** 8
**Confidence:** 4

**Summary:**

This paper considers the problem of out-of-domain generalization of discriminative models. The presented approach leverages domain relations and domain-specific meta-data to adapt to new domains at test time. Domain similarities are determined from a combination of a user-defined function and a learned function of the domain meta-data. The domain similarities are used to weight the different neural network heads in a mixture-of-experts fashion. At test time, the domain similarities are used to weight the output of all of the training set domain-specific heads to produce a prediction. Theoretical results provide justification for the proposed approach. Empirical results are shown on open source toy and real-world datasets, including several useful ablations.

**Strengths:**

* The motivation for the paper is strong, in a relevant area.
* The paper is well-written.
* The empirical results are very strong and well executed, including a large number of baselines. The ablation studies are beneficial, answering fundamental questions about the method.

**Weaknesses:**

* The method requires a user-defined domain similarity function based on meta-data. This might not always be available. Additionally, for specific applications, a user might not be able to credibly define such a function.
* The theoretical work makes (understandably) quite a few strong assumptions that are unlikely to be true in practice. These results could be improved to show how performance is affected by, for example, noisy domain relations.

**Questions:**

* Is it possible for the proposed method to utilize negative relationships between domains?
* How will the proposed method behave in a situation where there are no related domains according to the meta-data? Will the method perform as well as a domain-invariant approach? Is this what the consistency loss is for?
* If domain information was unavailable, could a domain discovery method (e.g. clustering) be used to learn domains and domain relations from unlabeled data? In the limit, could the model architecture and inference algorithm be adapted to learn domains and domain relations end-to-end?
* Could this method be used in data impoverished applications to improve performance?

Minor typos:
* Equations are referenced with "Eqn. equation #", maybe just "(#)" or "Equation (#)"?

---

> ### Author Response · Authors · 2023-11-21
> **Response to Reviewer 2jGa (1/3)**
>
> Thank you for reviewing our paper and for your valuable feedback. Below, we address your concerns point by point and we’ve revised our paper according to your suggestions. We would appreciate it if you could let us know whether your concerns are addressed by our response.
>
> **Q1**: Availability of domain meta-data and domain similarity function.
>
> **A1**: Most often, domain meta-data is obtained as a by-product during the collection of data, which then serves as the primary criteria for classifying domains. This meta-data typically manifests as an inherent property of the domain, such as latitude and longitude in geographic data or PPI scores in biomedical data.  Furthermore, obtaining meta-data for each testing domain is also not overly expensive since the number of testing domains is relatively small in most scenarios. As for the domain similarity function, we employ cosine similarity across all datasets, except in the case of biomedical data with PPI scores. Therefore, when we do not have specific metrics in certain fields, the alternative is to choose from a range of general distance metrics.
>
> In situations where domain metadata and the domain similarity function are unavailable, we can derive domain distances directly from the data. To do this, we compute the fixed relation using optimal transport [1]. For the learnable part, we utilize the average of features from each domain as input to the two-layer neural network $g$. We report the results in Table R1.
>
> **Table R1**: Performance comparison between D$^3$G and its variant that does not utilize domain meta-data, along with reporting the best baseline performance for each dataset.
>
> |      |   DG-15 (Avg. Acc. $\uparrow$)    |        TPT-48 (MSE $\downarrow$)        |                           | FMoW (Worst Acc. $\uparrow$) |              | ChEMBL-STRING (ROC-AUC $\uparrow$) |              |
> | :--------------- | :-----------------: | :-----------------------: | :-----------------------: | :-------------------: | :-------------------: | :--------------------: | :-------------------: |
> |                  |                     | N(24) $\rightarrow$ S(24) | E(24) $\rightarrow$ W(24) |       FMoW-Asia       |      FMoW-WILDS       |         PPI>50         |        PPI>100        |
> |         Best baseline        |   66.2 $\pm$ 2.0%   |     0.371 $\pm$ 0.054     |     0.262 $\pm$ 0.034     |   26.99 $\pm$ 1.27%   |   37.64 $\pm$ 0.92%   | 75.42 $\pm$ 0.42% | 73.57 $\pm$ 0.21% |
> |         D$^3$G w/o domain meta-data        |   70.2 $\pm$ 3.7%   |     0.381 $\pm$ 0.071     |     0.325 $\pm$ 0.029     |   27.17 $\pm$ 1.09%   |   37.82 $\pm$ 0.31%   |   76.65 $\pm$ 0.22%   |   75.01 $\pm$ 0.36%   |
> | **D$^3$G(Ours)** | **77.5 $\pm$ 2.5%** |   **0.342 $\pm$ 0.019**   |   **0.236 $\pm$ 0.063**   | **28.12 $\pm$ 0.28%** | **39.47 $\pm$ 0.57%** | **78.67 $\pm$ 0.16%**  | **77.24 $\pm$ 0.30%** |
>
> The findings indicate that this variant exhibits inferior performance compared to the one with domain meta-data, however, it still outperforms the best baselines on most datasets. These outcomes suggest that D$^3$G's model design inherently enhances generalization capabilities, which are further augmented by the use of domain meta-data. We’ve added these results in Appendix H.4 in the updated paper.
>
> ---
>
> **Q2**: The theoretical work makes (understandably) quite a few strong assumptions that are unlikely to be true in practice. These results could be improved to show how performance is affected by, for example, noisy domain relations.
>
> **A2**:  Thank you for your suggestion. Our results can be extended to noisy domain relations by incorporating technical tools from the error-in-variable literature. Specifically, in this situation, our assumption will be extended to $\|h^{(i)}-h^{(j)}\|_\infty$ $\le G\cdot$ $\|Z^{(i)}-Z^{(j)}\|$, while our estimated representation $\hat{Z}^{(i)}=Z^{(i)}+\epsilon_i$ includes a zero-mean sub-Gaussian noise term $\epsilon_i$. Then, we can use technical tools from error-in-variable literature to denoise it, we will leave it as a future work.
>
> We acknowledge that our theoretical analysis relies on some assumptions. Nevertheless, we still believe that our current theory provides valuable insights supporting the idea that utilizing good domain relations leads to better generalization. In fact, our proof suggests that using domain relations can help reduce variance while sacrificing a slight amount of bias, resulting in a more favorable bias-variance trade-off.

---

> ### Author Response · Authors · 2023-11-21
> **Response to Reviewer 2jGa (2/3)**
>
> **Q3**: Is it possible for the proposed method to utilize negative relationships between domains?
>
> **A3**: Yes, the model can handle and utilize negative domain relations. If our understanding is correct, for a pair of domains $i$ and $j$, the existence of negative domain relations implies that updating domain $j$'s corresponding parameters with data from domain $i$ would negatively impact the performance of domain $j$. In such cases, if the fixed relation indicates these negative relationships or if the learned relation does so, we will set the domain relation $a_{ij}=0$ to prevent information propagation between these two domains.
>
> ---
>
> **Q4**: How will the proposed method behave in a situation where there are no related domains according to the meta-data? Will the method perform as well as a domain-invariant approach? Is this what the consistency loss is for?
>
> **A4**: In situations where there are no related domains according to the meta-data, we will use a uniform domain relation, implying that all $a_ij$ are identical. During the training process, we emphasize the data from each head's corresponding domain with predictive loss, while applying uniform weights to other training domains in the calculation of consistency loss. During inference, our method performs comparably to a domain-invariant approach by averaging the outputs from each head. We analyze this situation in the first row of Table 3 in our paper, with the corresponding results also detailed in Table R2.
>
> **Table R2**: Performance comparison between D$^3$G and its variant with no related domains.
>
>
> |      |   FMoW (Worst Acc. $\uparrow$)    |                                     | ChEMBL-STRING (ROC-AUC $\uparrow$) |                       |
> |:-----------------|:----------:|:----------:|:---------:|:---------:|
> |                  | FMoW-Asia | FMoW-WILDS | PPI>50 |        PPI>100        |
> | D3G with no related domains |   26.93 $\pm$ 0.47%   |          35.32 $\pm$ 0.66%          |        76.17 $\pm$ 0.21%        |   73.38 $\pm$ 0.13%   |
> | **D3G (Ours)** | **28.12 $\pm$ 0.28%** |        **39.47 $\pm$ 0.57%**        |      **78.67 $\pm$ 0.16%**      | **77.24 $\pm$ 0.30%** |
>
> The results reveal that without using domain relation, we can slightly outperform ERM, yet significantly underperforms compared to our D$^3$G. This indicates the importance of using accurate domain relation to weight losses and predictions from different domains/heads.
>
> ---
>
> **Q5**: If domain information was unavailable, could a domain discovery method (e.g. clustering) be used to learn domains and domain relations from unlabeled data? In the limit, could the model architecture and inference algorithm be adapted to learn domains and domain relations end-to-end?
>
> **A5**: Yes, we do an ablation study for this question by clustering all training data into several groups, with the number of groups matching the actual number of domains. Each group is treated as a separate domain, and domain relations are established based on the distances between the clustering centers. The results, as presented in Table R3, show that by clustering data to learn domains, our method's variant surpasses ERM in 5 out of 7 settings, although still lags behind D$^3$G. This indicates that the model architecture and inference algorithm can somehow be adapted to learn domains and domain relations simultaneously in an end-to-end manner. Moreover, the appropriate use of domain meta-data and domain relation will then significantly improve the performance. We’ve added these results in Appendix H.4 in the updated paper.
>
> **Table R3**: Performance comparison between ERM, D$^3$G and its variant that does not utilize domain information.
>
> |      |   DG-15 (Avg. Acc. $\uparrow$)    |        TPT-48 (MSE $\downarrow$)        |                           | FMoW (Worst Acc. $\uparrow$) |              | ChEMBL-STRING (ROC-AUC $\uparrow$) |              |
> | :------------------------- | :-----------------: | :-----------------------: | :----------------------: | :-------------------: | :-------------------: | :--------------------: | :-------------------: |
> |                            |                     | N(24) $\rightarrow$ S(24) | E(24) $\rightarrow$W(24) |       FMoW-Asia       |      FMoW-WILDS       |         PPI>50         |        PPI>100        |
> | ERM                        |   44.0 $\pm$ 4.6%   |     0.445 $\pm$ 0.029     |    0.328 $\pm$ 0.033     |   26.05 $\pm$ 3.84%   |   34.87 $\pm$ 0.41%   |   74.11 $\pm$ 0.35%    |   71.91 $\pm$ 0.24%   |
> | D$^3$G w/o domain split information |   58.1 $\pm$ 4.1%   |     0.429 $\pm$ 0.055     |    0.340 $\pm$ 0.095     |   25.46 $\pm$ 1.35%   |   36.06 $\pm$ 0.39%   |   74.94 $\pm$ 0.79%    |   73.38 $\pm$ 0.91%   |
> | **D$^3$G(Ours)**              | **77.5 $\pm$ 2.5%** |   **0.342 $\pm$ 0.019**   |  **0.236 $\pm$ 0.063**   | **28.12 $\pm$ 0.28%** | **39.47 $\pm$ 0.57%** | **78.67 $\pm$ 0.16%**  | **77.24 $\pm$ 0.30%** |

---

> > ### Author Response · Authors · 2023-11-21
> > **Response to Reviewer 2jGa (3/3)**
> >
> > **Q6**: Could this method be used in data impoverished applications to improve performance?
> >
> > **A6**: Yes, our method can be applied effectively in data-scarce applications. The consistency loss is utilized to establish connections between domains and facilitate training on data insufficient domains. Throughout the training process, for all examples from domain $d$, we use these examples and employ the overall loss not only to update its domain-specific function $f^{(d)}$ but also to influence other domain-specific functions with the consistency loss. Consequently, the utilization of data from one domain effectively updates all parameters, implying that domain-specific models in data-scarce domains are optimized not only by their respective domain data but also by other domains, particularly those that are similar. For example, if training domain $d$ is more similar to training domain $d-1$ than to domain $d+1$, it will have a stronger influence on the optimization of the function $f^{(d-1)}$.
> >
> >
> >
> > ---
> >
> > **Q7**: Equations are referenced with "Eqn. equation #", maybe just "(#)" or "Equation (#)"?
> >
> > **A7**: Thanks for pointing that out. We’ve fixed these issues in the updated paper.
> >
> > ---
> >
> > **References**:
> >
> > [1] David Alvarez-Melis and Nicolò Fusi. Geometric dataset distances via optimal transport. ArXiv, abs/2002.02923, 2020.

---

> > > ### Comment · Reviewer_2jGa · 2023-11-21
> > >
> > > Thank you for your response. I will be keeping my score as it stands.

---

### Official Review · Reviewer_nUD5 · 2023-11-06

**Soundness:** 2 fair
**Presentation:** 3 good
**Contribution:** 2 fair
**Rating:** 6
**Confidence:** 4

**Summary:**

This paper attempts to solve the multi-source multi-target domain generalization problem using domain relations. The authors claim that existing single domain-invariant or multiple domain-specific models that leverage equal weights on all domains fail to capture appropriate domain-specific correlations. To tackle this problem, the authors extract domain relations from domain meta-data and design a relation-aware consistency regularizer to weight the training domain-specific functions. The weighted functions on source domains are then transferred to predictions on target domains. A theoretical analysis is provided to prove that the proposed approach can perform better than the models using equal domain weights.

**Strengths:**

1. This paper is clearly written and organized. The underlying idea is straightforward and easy to follow.

2. The authors provide a theoretical analysis that verifies that the proposed relation-aware consistency loss can achieve superior generalization performance compared to the approach of treating all training domains
equally.

3. The experimental evaluation is comprehensive and thorough. An illustrative toy task is designed to examine the proposed methods and many tables and figures are presented to analyze the results.

**Weaknesses:**

1. The novelty of the proposed method may be somewhat limited as it bears similarities to ensemble methods, which also involve assigning varying weights to different domains. The essential distinctions between the proposed methods and ensemble methods should be clearly elucidated, and it is important to consider baseline models that incorporate ensemble methods for comparison.

2. The evaluation metrics used in this paper are different from previous works conducted on the TPT-48, FMoW, and ChEMBL-STRING datasets. It is unclear whether the metrics used, such as MSE, are suitable and fair to be used to evaluate domain generalization models.

3. As stated by the authors, the theoretical analysis of this paper relies on certain assumptions. It is questioning whether these assumptions reflect the real-world datasets. For example, the assumptions that the domain relations accurately capture the similarity between domains, and that they are determined solely by the distance between domain representations, may not always hold true in real-world datasets. In the proofs of the theorems, the authors only prove that the proposed estimator outperforms the equally weighted estimator in the minimax sense, but not a theoretically global superiority. Moreover, the theorems only reveal that uneven weights can be better than equal weights, which does not entirely support the key ideas of acquiring appropriate weights for different domains.

**Questions:**

Please see the weakness points.

---

> ### Author Response · Authors · 2023-11-21
> **Response to Reviewer nUD5 (1/2)**
>
> Thank you so much for the insightful and valuable comments! They are very helpful for further improving the clarity and quality of our paper. We'll revise our manuscript in the next version to address all of your concerns.
>
> **Q1**: Comparison between D$^3$G and ensemble methods.
>
> **A1**: We acknowledge that our proposed method shares some similarities with ensemble methods, yet it is more closely aligned with multi-head network structures. As illustrated in Figure 1 in the paper, our approach employs a shared feature extractor and learns a collection of domain-specific heads. Consequently, in our paper, we compare several multi-head models without using domain meta-data, such as DRM [1], LLE [2], and DDN [3]. Our method consistently outperforms these baselines, highlighting the significant role of domain meta-data in generating accurate domain relations and enhancing generalization performance.
>
> To further address your concern, we have also conducted an additional comparison between D$^3$G and one variant, which employs separate feature extractors for each training domain. The results are reported in Table R1 here.
>
> **Table R1**: Performance comparison between D$^3$G and using separate feature extractors for each training domain.
>
> |      |   DG-15 (Avg. Acc. $\uparrow$)    |        TPT-48 (MSE $\downarrow$)        |                           | FMoW (Worst Acc. $\uparrow$) |              | ChEMBL-STRING (ROC-AUC $\uparrow$) |              |
> | :--------------- | :-----------------: | :-----------------------: | :-----------------------: | :-------------------: | :-------------------: | :--------------------: | :-------------------: |
> |                  |                     | N(24) $\rightarrow$ S(24) | E(24) $\rightarrow$ W(24) |       FMoW-Asia       |      FMoW-WILDS       |         PPI>50         |        PPI>100        |
> |         Using separate feature extractors        |   47.3 $\pm$ 3.7%   |     0.563 $\pm$ 0.071     |     0.519 $\pm$ 0.076     |   23.62 $\pm$ 0.47%   |   35.44 $\pm$ 0.58%   |   76.11 $\pm$ 0.31%    |   74.19 $\pm$ 0.74%   |
> | **D$^3$G (Ours)** | **77.5 $\pm$ 2.5%** |   **0.342 $\pm$ 0.019**   |   **0.236 $\pm$ 0.063**   | **28.12 $\pm$ 0.28%** | **39.47 $\pm$ 0.57%** | **78.67 $\pm$ 0.16%**  | **77.24 $\pm$ 0.30%** |
>
> According to the results, D$^3$G shows superior performance compared to using separate feature extractors for each training domain. This underscores the importance of employing a shared feature extractor to learn a universal representation, while domain-specific heads can further identify domain-specific features. Additionally, our D$^3$G model demonstrates superior efficiency compared to this variant since it utilizes a shared feature extractor while the variant needs $N^{tr}$ feature extractors. We’ve added these results in Appendix H.3 in the updated paper.
>
> ---
>
> **Q2**: Evaluation metrics on the TPT-48, FMoW, and ChEMBL-STRING datasets
>
> **A2**: In this paper, the evaluation metrics we use are those originally selected by the creators of the respective datasets:
> - TPT-48: we follow GRDA [4] and use MSE as our evaluation metric.
> - FMoW: we use worse accuracy as our evaluation metric following Wilds [5].
> - ChEMBL-STRING: we use ROC-AUC as our evaluation metric following SGNN-EBM [6].
>
> ---
>
> **Q3**: As stated by the authors, the theoretical analysis of this paper relies on certain assumptions. It is questioning whether these assumptions reflect the real-world datasets. In the proofs of the theorems, the authors only prove that the proposed estimator outperforms the equally weighted estimator in the minimax sense, but not a theoretically global superiority. Moreover, the theorems only reveal that uneven weights can be better than equal weights, which does not entirely support the key ideas of acquiring appropriate weights for different domains.
>
> **A3**: Thank you for your comment. We acknowledge that our theoretical analysis relies on some assumptions. The aim of our theoretical analysis is to explain some phenomena distilled from our empirical study and to help us understand why D$^3$G can work. In the future, we plan to relax these assumptions.
>
> Furthermore, we would like to note that the comparison in the minimax sense is a common approach in machine learning and statistics literature. For example, you can find it in papers, such as meta-learning [7], reinforcement learning [8], and private learning [9, 10]. Regarding your second point about uneven weights, we want to address it with care. It appears there might be a potential misunderstanding. In our theory, we propose that weighting according to similarity (controlled by the parameter $B$) can yield better results than using equal weights, as opposed to using uneven weights.

---

> ### Author Response · Authors · 2023-11-21
> **Response to Reviewer nUD5 (2/2)**
>
> **References**:
>
> [1] Zhang, Yi-Fan, Jindong Wang, Jian Liang, Zhang Zhang, Baosheng Yu, Liang Wang, Dacheng Tao, and Xing Xie. "Domain-Specific Risk Minimization for Domain Generalization." In Proceedings of the 29th ACM SIGKDD Conference on Knowledge Discovery and Data Mining, pp. 3409-3421. 2023.
>
> [2] Zhiheng Li, Ivan Evtimov, Albert Gordo, Caner Hazirbas, Tal Hassner, Cristian Canton Ferrer, Chenliang Xu, and Mark Ibrahim. A whac-a-mole dilemma: Shortcuts come in multiples where mitigating one amplifies others. June 2023. URL https://arxiv.org/abs/2212.04825.
>
> [3] Daoan Zhang, Mingkai Chen, Chenming Li, Lingyun Huang, and Jianguo Zhang. Aggregation of disentanglement: Reconsidering domain variations in domain generalization. ArXiv, abs/2302.02350, 2023.
>
> [4] Zihao Xu, Hao He, Guang-He Lee, Yuyang Wang, and Hao Wang. Graph-relational domain adaptation. arXiv preprint:2202.03628, 2022.
>
> [5] Pang Wei Koh, Shiori Sagawa, Sang Michael Xie, Marvin Zhang, Akshay Balsubramani, Weihua Hu, Michihiro Yasunaga, Richard Lanas Phillips, Irena Gao, Tony Lee, et al. Wilds: A benchmark of in-the-wild distribution shifts. In International Conference on Machine Learning, pages 5637–5664. PMLR, 2021b.
>
> [6] Shengchao Liu, Meng Qu, Zuobai Zhang, Huiyu Cai, and Jian Tang. Structured multi-task learning for molecular property prediction. In International Conference on Artificial Intelligence and Statistics, pages 8906–8920. PMLR, 2022.
>
> [7] Tripuraneni, Nilesh, Chi Jin, and Michael Jordan. "Provable meta-learning of linear representations." In International Conference on Machine Learning, pp. 10434-10443. PMLR, 2021.
>
> [8] Li, Gen, Yuling Yan, Yuxin Chen, and Jianqing Fan. "Minimax-optimal reward-agnostic exploration in reinforcement learning." arXiv preprint arXiv:2304.07278 (2023).
>
> [9] Duchi, John C., Michael I. Jordan, and Martin J. Wainwright. "Minimax optimal procedures for locally private estimation." Journal of the American Statistical Association 113, no. 521 (2018): 182-201.
>
> [10] Asi, Hilal, and John C. Duchi. "Near instance-optimality in differential privacy." arXiv preprint arXiv:2005.10630 (2020).

---

> ### Comment · Reviewer_nUD5 · 2023-11-22
>
> Thanks for your detailed response. I will be keeping my rating.

---

### Meta-Review · Area_Chair_SmxA · 2023-12-07

**Metareview:**

(a) The paper addresses the multi-source multi-target domain generalization problem by leveraging domain relations. The proposed method extracts domain relations from metadata, uses a relation-aware consistency regularizer to weight domain-specific functions, and transfers these weighted functions to target domains. Theoretical analysis supports the claim that this approach outperforms models with equal domain weights.

(b) Strengths include clear organization, a straightforward underlying idea, comprehensive experimental evaluation, and a theoretical analysis validating the proposed relation-aware consistency loss. The paper is well-written, and the empirical results are thorough, featuring illustrative toy tasks and numerous tables and figures.

(c) Weaknesses encompass potential novelty limitations resembling ensemble methods, differences in evaluation metrics from previous works, and reliance on assumptions in theoretical analysis. Theoretical proofs focus on minimax superiority, and concerns arise about the accuracy of assumptions in real-world datasets. Clarification on distinctions from ensemble methods, consideration of baseline models, and addressing evaluation metric differences are suggested improvements.

**Justification For Why Not Higher Score:**

N/A

**Justification For Why Not Lower Score:**

The reviewers highlighted several strengths of the paper, including its clarity, theoretical analysis, and strong empirical results. They also pointed out potential weaknesses such as similarity to ensemble methods, theoretical assumptions, and the clarity around certain methodological aspects. The main justification for not giving a lower score might stem from the paper’s well-executed experiments and the potential significance of the proposed method despite its limitations. The method introduces a novel approach in dealing with domain relations, though it might need further clarification and differentiation from existing techniques.

---

### Decision · Program_Chairs · 2024-01-16

Accept (spotlight)